# Human-induced intensified seasonal cycle of sea surface temperature

Fukai Liu [1] ✉, Fengfei Song [1,2] ✉ & Yiyong Luo [1] ✉

Changes in the seasonal cycle of sea surface temperature (SST) have far-reaching ecological and societal implications. Previous studies have found an intensified SST seasonal cycle under global warming, but whether such changes have emerged in historical records remains largely unknown. Here, we reveal that the SST seasonal cycle globally has intensified by $3.9 \pm 1.6\%$ in recent four decades (1983–2022), with hotspot regions such as the northern subpolar gyres experiencing an intensification of up to 10%. Increased greenhouse gases are the primary driver of this intensification, and decreased anthropogenic aerosols also contribute. These changes in anthropogenic emissions lead to shallower mixed layer depths, reducing the thermal inertia of upper ocean and enhancing the seasonality of SST. In addition, the direct impacts of increased ocean heat uptake and suppressed seasonal amplitude of surface heat flux also contribute in the North Pacific and North Atlantic. The temperature seasonal cycle is intensified not only at the ocean surface, but throughout the mixed layer. The ramifications of this intensified SST seasonal cycle extend to the seasonal variation in upper-ocean oxygenation, a critical factor for most ocean ecosystems.

Sea surface temperature (SST) plays a pivotal role in linking the atmosphere and ocean, attracting extensive attention to deciphering variations in annual and seasonal mean SST[1–3]. However, even in the absence of annual mean state changes, SST retains the potential to exert substantial climatic influence by modulating the amplitude of its seasonal cycle. These changes in the SST seasonal cycle can have profound climatic impacts, influencing phenomena such as marine heat waves, Asian monsoons, precipitation, and the El Niño-Southern Oscillation[4–7]. Furthermore, changes in SST seasonality bear considerable ecological implications, particularly concerning oceanic oxygen content, a factor directly influencing ocean productivity and biogeochemical cycles[8–12].

In a warmer future climate, climate models consistently project a global intensification of the SST seasonal cycle[13–17]. Under extreme warming scenarios, Coupled Model Intercomparison Project phase 5 (CMIP5) models indicate that the amplitude of the SST seasonal cycle could intensify by ~30% by the end of the 21st century compared to the pre-industrial levels[14]. The precise mechanisms propelling this

intensification remain a subject of debate. Some studies have emphasized the thermodynamic effects of atmospheric circulation changes, which modify the seasonal cycle of surface heat fluxes and subsequently impact the seasonal cycle of SST[18–20]. Other studies have highlighted the crucial role of changes in the mixed layer depth (MLD) in driving the intensified SST amplitude[13,14,17]. As the Earth warms, increased ocean heat uptake leads to enhanced stratification in the upper ocean, resulting in a shallower MLD[21–23]. The reduced heat content within the shallower MLD diminishes thermal inertia, intensifying the SST's seasonal cycle.

Despite the consensus on the intensified SST seasonal cycle in future warming scenarios, it remains challenging to determine whether such changes have already manifested themselves in the observations. Unlike future projections, the sparse observational record and the influence of internal variability make it much more difficult to detect these changes in the observational period. In addition, existing studies that have explored the detection of seasonal cycle changes in surface temperature have focused primarily on land[24–28], leaving the emergence

[1]Frontiers Science Center for Deep Ocean Multispheres and Earth System and Physical Oceanography Laboratory, Ocean University of China, Qingdao, China.
[2]Laoshan Laboratory, Qingdao, China. ✉e-mail: fliu@ouc.edu.cn; songfengfei@ouc.edu.cn; yiyongluo@ouc.edu.cn

of surface temperature trends over the oceans largely elusive. Our study fills this gap by using a combination of observations and model simulations, unveiling a substantial enhancement of the SST seasonal cycle over most of the oceans during the past four decades.

## Results

### Intensified global SST seasonal cycle in observations

The amplitude of the SST seasonal cycle is assessed by calculating the difference between the maximum and minimum values of the annual cycle at each grid point (see "Amplitude of seasonal cycle" in Methods). For our investigation, we rely primarily on three widely-used observational SST datasets: the Extended Reconstructed SST dataset (ERSST) version 5, the Hadley Centre Sea Ice and SST dataset (HadISST) version 1.1, and the high-resolution Optimum Interpolation Sea Surface Temperature (OISST) version 2. These datasets cover the period from 1982 to 2022, enabling a detailed examination of the SST seasonal

cycle over the past four decades (see "Observational and reanalysis datasets" in Methods). Pre-1982 observations are excluded from our analysis due to the sparse observations, particularly during the deep winter season in high latitudes[29], which may not adequately capture the SST seasonal cycle. We denote the mean of the three datasets as OBS.

Climatologically, the Northern Hemisphere SST in OBS exhibits a maximum in August and a minimum in February (Fig. 1a), while the Southern Hemisphere shows the opposite evolution (Supplementary Fig. 1). Over the past four decades, the main changes in SST seasonality are an enhanced difference between the annual maximum and minimum temperature (Fig. 1a, bars), demonstrating a substantial increase in amplitude. The intensified SST seasonal cycle is further confirmed by the time series of the seasonal cycle amplitude in all three observations (Fig. 1c, black lines). Overall, the intensification in OBS corresponds to a statistically significant increase of 0.16 ± 0.07 °C (the

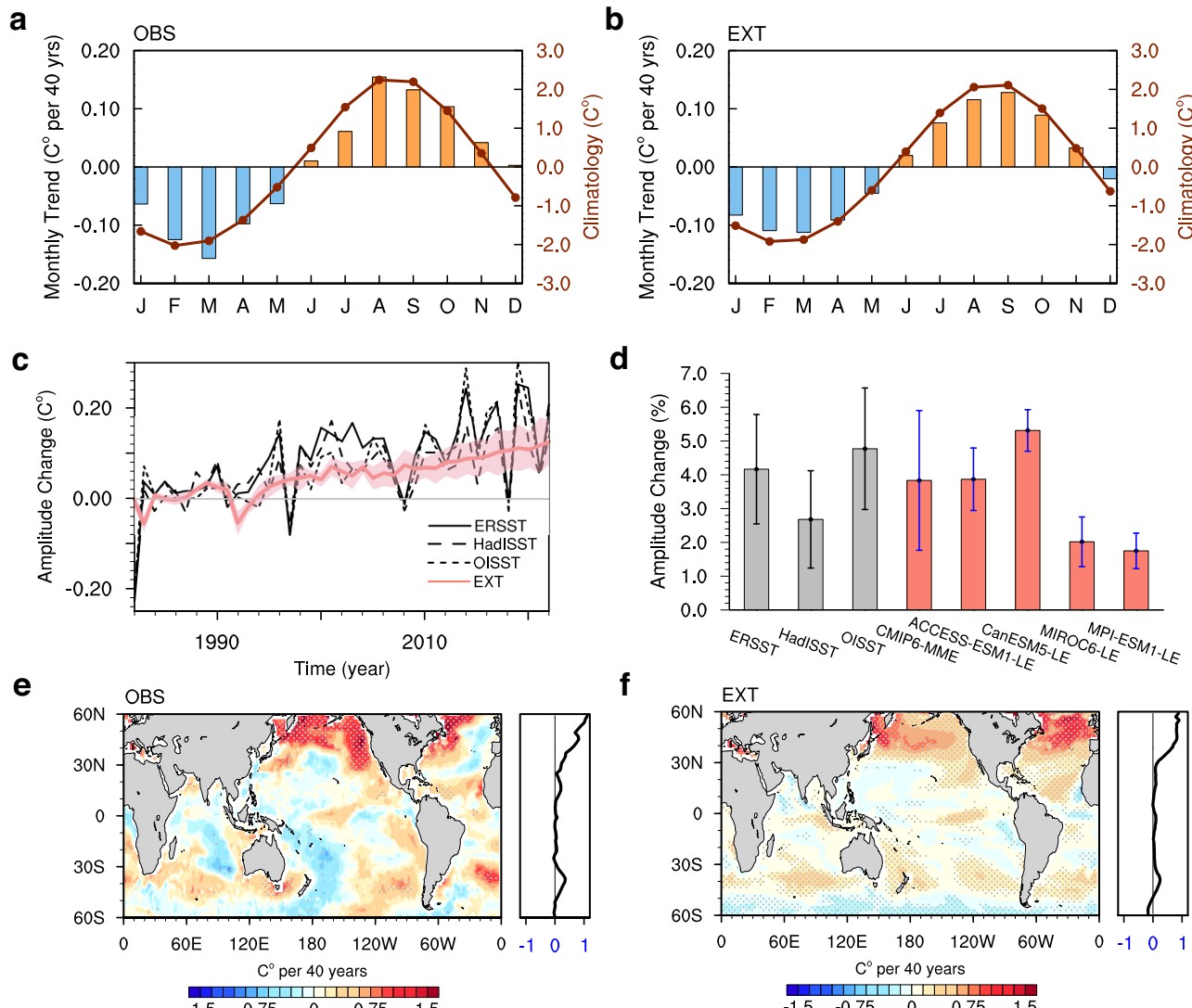

**Fig. 1 | Observed and simulated intensification in sea surface temperature (SST) seasonal cycle. a, b**, The seasonal cycle of SST averaged over the Northern Hemisphere (dotted line; 0–60 °N) and its trend (bars) during the period 1983–2022 from (**a**) observations (OBS; mean of ERSST, HadISST, and OISST) and (**b**) external forcing simulation ensembles (EXT; mean of CMIP6 multimodel ensemble (MME), ACCESS-ESM1-5 large ensemble (LENS), CanESM5 LENS, MIROC6 LENS, and MPI-ESM1-2-LR LENS). **c** The temporal evolution of the global mean amplitude of SST seasonal cycle (unit: °C) from observations (black lines), and EXT model simulations from the historical (1982–2014) and SSP5-8.5 scenarios

(2015–2022) (red line) with standard deviation among five simulation ensembles (pink shading). **d** Linear trends of the global mean amplitude of SST seasonal cycle during 1983–2022 relative to the climatological mean of 1983–1992 (unit: %). Error bars represent the 5–95% confidence levels associated with trends for each observation and one standard deviation among ensemble members for each model simulation. **e, f** Linear trends of SST seasonal cycle amplitude (unit: °C per 40 y) during 1983–2022 from (**e**) OBS and (**f**) EXT, with their zonal means shown in the right-hand panels. Stippling indicates where the trend is statistically significant above the 95% confidence level based on Student's *t* test.

confidence interval is 5–95%) from 1983 to 2022, representing a relative increase of 3.9 ± 1.6% compared to the climatological mean of 4.2 °C during 1983–1992 (Fig. 1d, gray bars). Note that the first year, 1982, is excluded from the trend analysis due to its extremely low value, which would skew the calculated trend.

To tease out the influence of internal variability and to derive an estimate of the forced response of the SST seasonal cycle to external forcing, we average the results of five simulation ensembles, comprising a total of 181 individual realizations (denoted as "EXT"; see "Climate model simulations" in Methods). The EXT ensemble successfully reproduces the observed seasonal cycle (Fig. 1b) and its amplification (Fig. 1c, red line), with an increase of ~3.4% over the period from 1983 to 2022. The intensification trend is significantly robust across all five simulation ensembles (Fig. 1d, red bars), ranging from ~1.7% in MPI-ESM1-LE to ~5.3% in CanESM5-LE. Moreover, this intensification trend appears in 35 out of 36 CMIP6 models, with intensifications ranging from 0.3% in GISS-E2-2-G to 7.0% in IPSL-CM6A-LR (Supplementary Fig. 2). The high consistency of global mean amplitude trends between observations and model simulations indicates that the intensification is primarily a robust climate response to external forcing, rather than being internally generated.

The linear trend of the observed amplitude between 1983 and 2022 displays considerable regional variations (Fig. 1e). The most substantial intensification occurs in the northern subpolar gyres, reaching ~11.9% between 45°N and 60°N. In the North Pacific, a distinctive horseshoe-shaped pattern of seasonal cycle enhancement emerges, while in the North Atlantic, the intensification primarily occurs in the western basin. In the Southern Hemisphere, prominent and extensive intensification (~5.0% between 35°S and 50°S) is observed north of the Atlantic Circumpolar Circulation (ACC). Intensifications also appear in the eastern tropical Pacific and northeast tropical Atlantic. However, specific regions exhibit a significant reduction in amplitude, including extensive areas of the western Pacific Ocean and the Indian Ocean, as well as the southern boundaries of the ACC. This leads to minor changes in the zonal mean amplitude near 30°N and 30°S. The pattern of amplitude change is consistent across the ERSST, HadISST, and OISST datasets (Supplementary Fig. 3).

By prescribing the external forcings, the EXT can well capture the major spatial features of observed changes in seasonal cycle amplitude, with a pattern correlation of 0.56. Positive trends are present in the North Pacific, North Atlantic, and the northern flank of the ACC, while muted and negative trends occur in the southern flank of the ACC, the western subtropical Pacific (comparing Fig. 1f, e). When considering only statistically significant trends in OBS (stippled regions in Fig. 1e), the pattern correlation increases to 0.85. This spatial resemblance between OBS and simulations remains consistent across all five simulation ensembles (Supplementary Fig. 4). Notably, discrepancies between observed and simulated amplitude changes are more pronounced in the Southern Hemisphere, particularly in the Pacific sector of the Southern Ocean. The pattern correlation is substantially lower in the Southern Hemisphere (0°–60°S) at 0.24 compared to 0.72 in the Northern Hemisphere (0°–60°N), suggesting the influence of large observational uncertainties in the data-sparse Southern Ocean. Furthermore, the low-frequency internal variability within the climate system[30] may also contribute, as most of the discrepancy between EXT and OBS falls within the range of CMIP6 ensemble standard deviation (Supplementary Fig. 4b). Despite these discrepancies, the model simulations effectively replicate the temporal evolution and the magnitude of intensification trends of SST seasonal cycle in hotspot regions, thus highlighting the dominant role of external forcing.

## Attribution of the observed trend in the SST seasonal cycle
To discern the effects of different external forcings, we employ model simulations from the Detection and Attribution Model

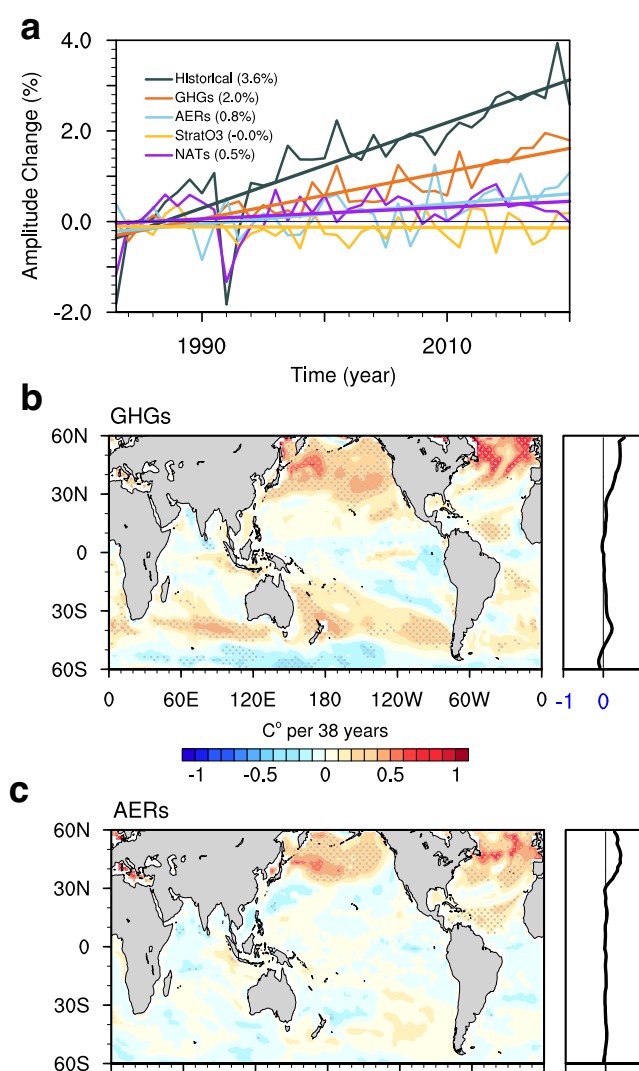

**Fig. 2 | Attribution of sea surface temperature (SST) seasonal cycle amplitude to different radiative forcings. a** Temporal evolution of changes in the global mean amplitude of SST seasonal cycle (unit: %) in ensembled historical, greenhouse gases (GHGs), anthropogenic aerosols (AERs), stratospheric ozone (StratO3), and natural forcings (NATs) simulations, relative to the climatological mean of 1983–1992. Linear trends during 1983–2020 are also indicated in parentheses. **b**, **c** Linear trend of SST seasonal cycle amplitude (unit: °C per 38 y) in (**b**) GHGs and (**c**) AERs during 1983–2020, with their zonal means (unit: % per 38 y) shown in the right-hand panels. Stippling indicates where the trend is statistically significant above the 95% confidence level based on Student's *t*-test.

Intercomparison Project (DAMIP) (see "Climate model simulations" in Methods), which enables separating the contributions of four major radiative forcings: greenhouse gases (GHGs), anthropogenic aerosols (AERs), stratospheric ozone (StratO3), and natural forcings (NATs) associated with volcanic and solar variations.

During 1982–2020, the continuous increase in GHGs concentration has amplified the SST seasonal cycle by approximately ~2.0% (Fig. 2a, red line), accounting for ~55.6% of the total historical enhancement in CMIP6 models (Fig. 2a, black line). The AERs also play a role in amplifying SST seasonality by ~0.8% (Fig. 2a, light blue line), as a result of subsequent reduction in aerosol emissions in Europe, North America, and China since the 1980s[31]. In contrast, contributions from

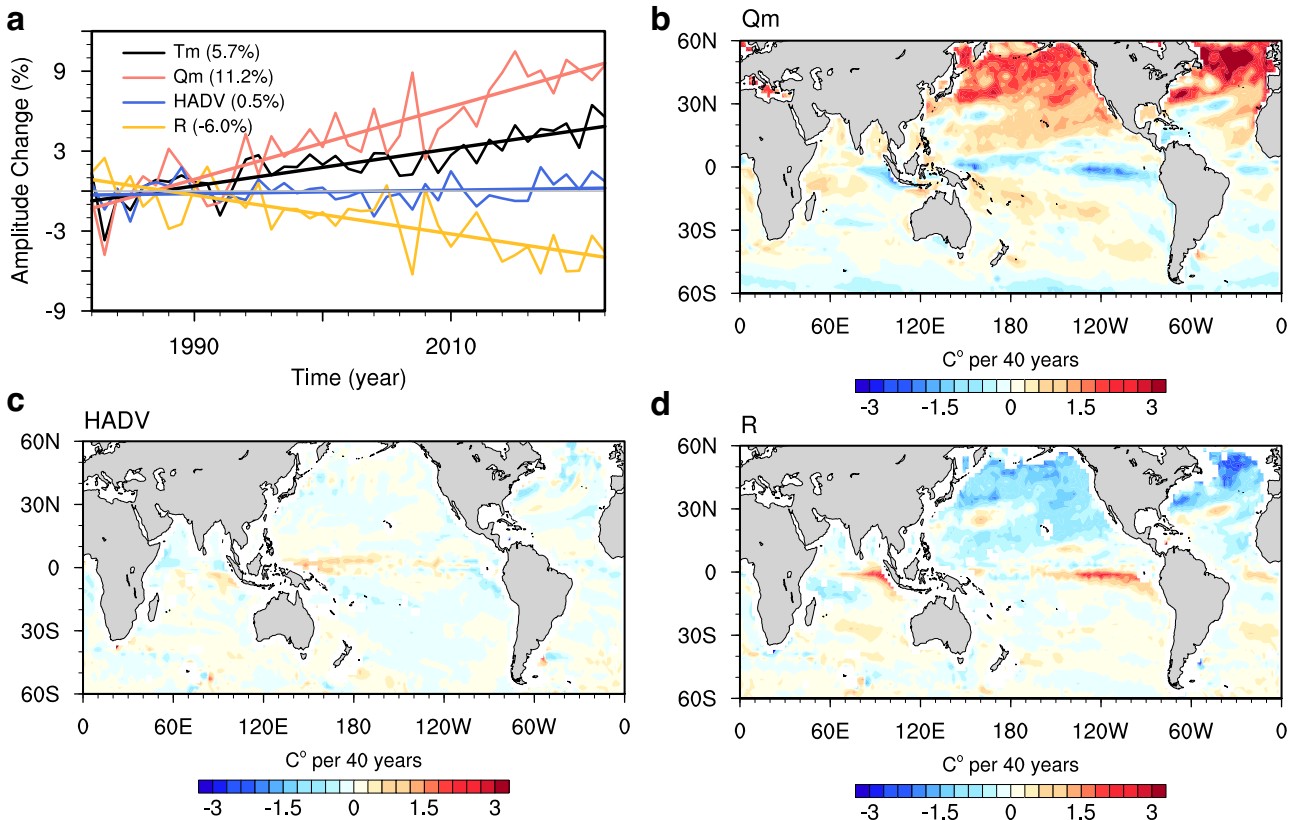

**Fig. 3 | Causes of intensification in sea surface temperature (SST) seasonal cycle. a** Temporal evolution of changes in the global mean amplitude of mixed layer temperature (Tm; black line) seasonal cycle (unit: %) from CMIP6 multimodel ensemble relative to the mean of 1983–1992, and its decomposition into contributions from the thermal forcing term (Qm; red line), the horizontal advection term (HADV; blue line), and the residual term (R; orange line). **b–d** Linear trend of SST seasonal cycle amplitude during 1983–2022 (unit: °C per 40 y) due to (**b**) the thermal forcing term, (**c**) the horizontal advection term, and (**d**) the residual term from CMIP6 multimodel ensemble.

StratO3 have barely any contribution to the seasonal amplitude changes (Fig. 2a, yellow line). The ~0.4% intensification caused by NATs largely results from two abrupt drops occurring in 1983 and 1992 and thus is not statistically significant (Fig. 2a, purple line). These drops follow the explosive volcanic eruptions of El Chichon in 1982 and Mount Pinatubo in 1991, respectively, underscoring the discrete impact of NATs on specific periods of the SST seasonal cycle.

Similar to its impact on temporal evolution, GHGs are the primary driver shaping the spatial pattern of amplitude changes in the historical simulations (comparing Fig. 2b to Fig. 1f), including the widespread intensification in the northern subpolar gyres and the dipole-like trends in the Southern Ocean. However, there are some differences between historical simulations and GHGs simulations. In the historical simulations, the intensification in the North Pacific is stronger and more concentrated within the subpolar gyre, whereas GHGs-induced intensification is weaker and spreads more into lower latitudes. The inclusion of AERs offers a plausible explanation for the differences between GHGs and historical simulations. For example, AERs generate an amplitude dipole in the North Pacific centered at ~35°N (Fig. 2c), where the amplitude increases on the poleward side and decreases on the equatorward side. This dipole structure contributes to the meridional contrast in the North Pacific in historical simulations, strongly enhancing the intensification in the subpolar regions. Overall, it is evident that the intensification of the SST seasonal cycle is predominantly driven by human-induced GHGs emissions, and AERs also play a significant role in northern subpolar oceans.

Climate models robustly capture the observed intensification of the SST seasonal cycle, adding confidence to their future projections (Supplementary Fig. 5). In the high-emissions Shared Socioeconomic

Pathway scenario 5–8.5 (SSP5-8.5), the SST seasonal cycle exhibits an almost linear upward trend, resulting in a remarkable ~10.6% enhancement by 2100 compared to the present-day level (mean of 2015–2024). Even under lower emission scenarios like SSP2-4.5 and SSP3-7.0, the intensification can reach ~4.8% and ~7.7%, respectively. It is important to note that certain ocean areas experience more considerable intensification due to regional variations. For instance, in the North Pacific and North Atlantic, the amplitude will enhance by ~10.3% between 45°N and 60°N, even in the modest SSP2-4.5 scenario (Supplementary Fig. 5b).

## Mechanisms of the intensified SST seasonal cycle

To explore the mechanisms driving changes in the SST seasonal cycle over the past four decades, we utilize CMIP6 simulations and conduct a mixed layer budget analysis (see "Mixed Layer Heat Budget" in Methods). This approach enabled us to quantitatively assess the contributions of different factors (Figs. 3, 4), including the thermal forcing term $Q_m$ that is proportional to the ratio between the net surface heat flux (SHF) and MLD, as well as horizontal advection and other residual processes.

During 1983–2022, $Q_m$ exhibits positive trends across most of the global ocean (Fig. 3b), particularly noteworthy in the northern subpolar gyres, where induced seasonal intensification reaches 1.8 °C north of 30°N, corresponding to a 58.1% increase relative to the 1983–1992 mean. The residual term emerges as the primary contributor to seasonal intensification in the equatorial oceans and also plays a role in the intensification on the northern flank of the ACC (Fig. 3d), and it strongly suppresses the seasonal amplitude in the Northern Hemisphere, indicating the importance of vertical

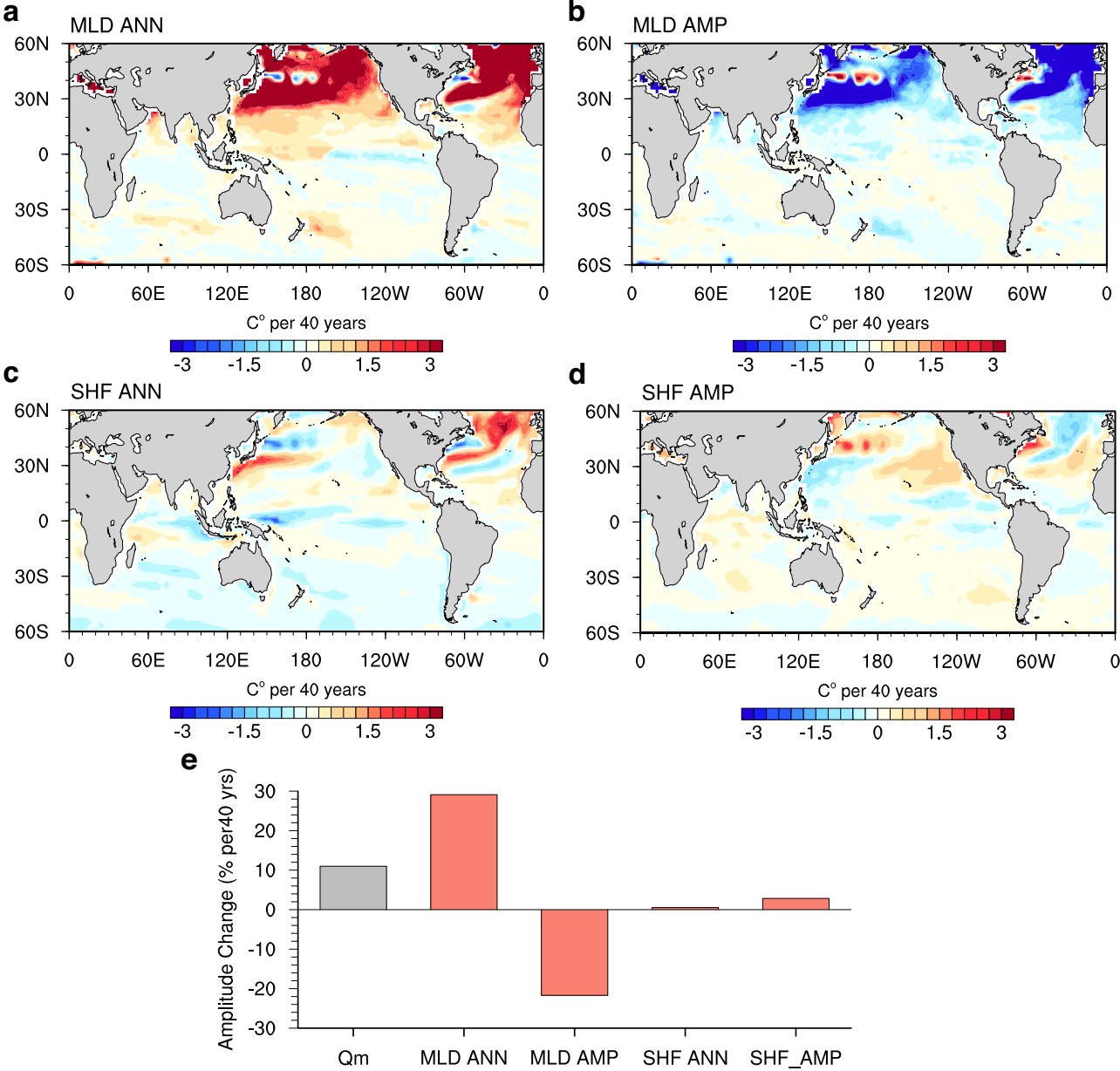

**Fig. 4 | Dominance of shoaling mixed layer in amplified sea surface temperature (SST) seasonal cycle. a–d** Linear trends of the mixed layer temperature seasonal amplitude during 1983–2022 from CMIP6 multimodel ensemble (unit: °C per 40 y) due to changes in (**a**) annual mean mixed layer depth (MLD ANN), (**b**) seasonal amplitude of mixed layer depth (MLD AMP), (**c**) annual mean surface heat flux (SHF ANN), and (**d**) seasonal amplitude of surface heat flux (SHF AMP). **e** Linear trends of the global-mean mixed layer temperature amplitude due to thermal forcing (Qm; gray bar), as well as its decomposition into different contributing factors (red bars).

entrainment or diffusive processes in these regions. In the global mean (Fig. 3a), the amplification of the SST seasonal cycle is predominantly attributed to positive contributions from $Q_m$ (-11.2%; red), and this effect is partially counteracted by the residual term (−−6.0%; orange). The contribution from the horizontal advection is deemed negligible (blue).

This investigation naturally prompts an inquiry into the primary factors shaping $Q_m$. These factors include decreased MLD (Fig. 5a), reduced seasonal amplitude of MLD (Supplementary Fig. 6a), increased ocean heat uptake (Supplementary Fig. 6b), and enhanced amplitude of SHF across most of the global oceans (Supplementary Fig. 6c). Through detailed decomposition (Fig. 4; see "Mixed Layer Heat Budget" in Methods), our analysis highlights the predominant role of shallower MLD over the past four decades, largely attributed to enhanced upper ocean stratification[32–35]. The impact of decreased MLD

on SST seasonality is most prominent in the northern subpolar oceans, where the induced seasonal intensification exhibits a remarkable 158.6% increase relative to the 1983–1992 mean. In the Southern Hemisphere, although the effect of MLD reduction is relatively smaller, it still contributes to an intensification around 35°S. Globally, the decreased MLD entirely drives the evolution in $Q_m$, amplifying the SST seasonal cycle by 29.1% (Fig. 4e).

The intensification of the SST seasonal cycle due to decreased MLD is largely offset by the seasonal amplitude change in MLD (Fig. 4b), the latter is characterized by a broad reduction in mid- to high-latitude oceans (Supplementary Fig. 6a). The underlying mechanisms are depicted schematically in Supplementary Fig. 7: considering a scenario where SHF oscillates annually around zero and MLD also follows an annual-repeating evolution, albeit with reduced seasonal amplitude and no long-term trend. This results in a $Q_m$

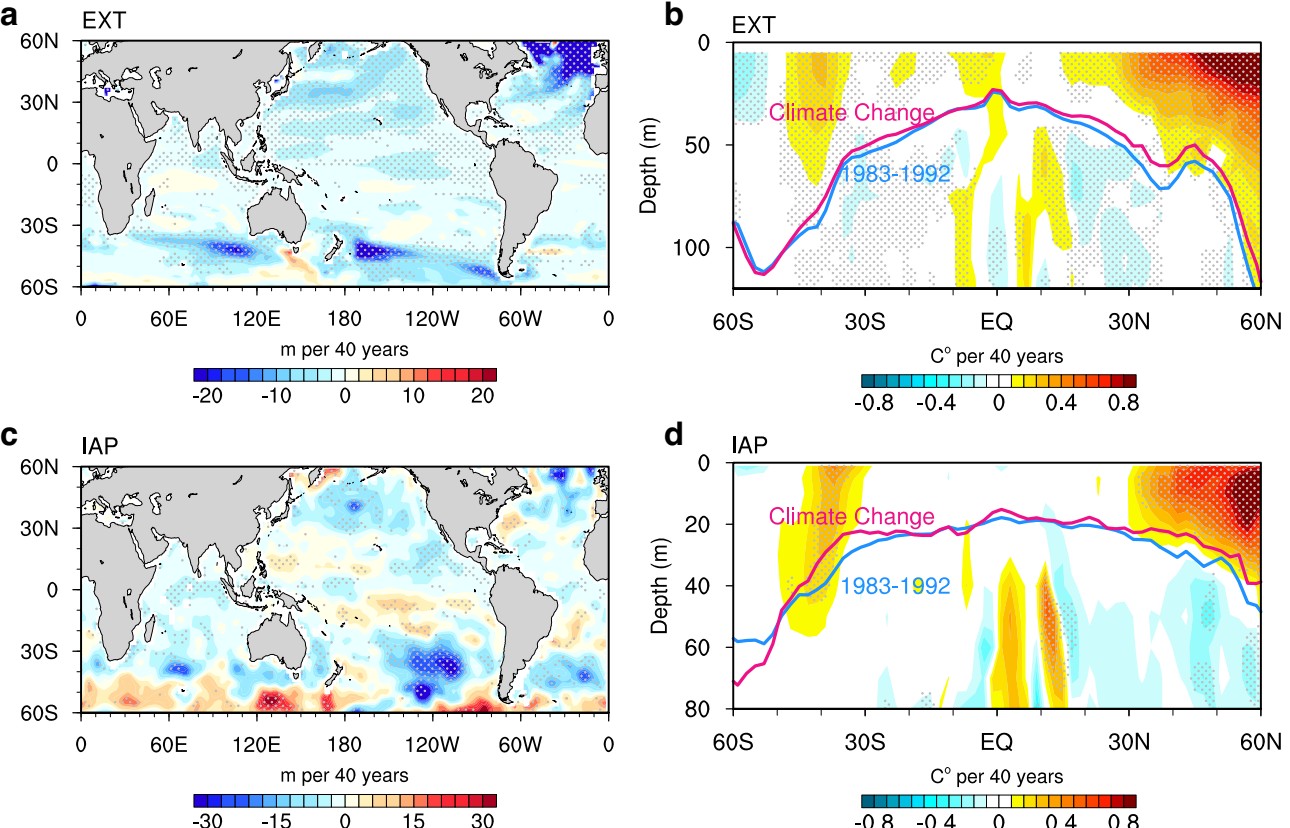

**Fig. 5 | Intensification of temperature seasonal cycle across the mixed layer.**
**a** Linear trend of annual-mean mixed layer depth (MLD; unit: m per 40 y) during 1983–2022 from the external forcing simulation ensembles (EXT; mean of CMIP6 multimodel ensemble (MME), ACCESS-ESM1-5 large ensemble, CanESM5 large ensemble, and MPI-ESM1-2-LR large ensemble). **b** Linear trend of the zonal mean seasonal cycle of temperature (unit: °C per 40 y) during 1983–2022 from the EXT (mean of CMIP6 MME). The blue and red lines denote the mean MLD averaged over 1983–1992 and its response (the mean over 1983–1992 plus the trend over 1983–2022), respectively. **c**, **d** Same as (**a**, **b**), but for the IAP dataset. Stippling indicates where the trend is statistically significant above the 95% confidence level based on Student's *t* test.

characterized by decreased maxima in summer and increased minima in winter. However, the rate of decrease during summer surpasses the rate of increase in winter, owing to shallower MLD in warmer seasons. Consequently, the diminished seasonal cycle of MLD substantially contributes to a reduction in the SST seasonal cycle amplitude, effectively mitigating the contribution from decreasing MLD.

Changes in the annual mean and seasonal amplitude of SHF contribute to regional intensifications of SST seasonality (Fig. 4c, d): the anomalous heat uptake over the North Atlantic and around the Kuroshio (Supplementary Fig. 6b) intensify the SST seasonal cycle there, while increased amplitude of SHF over the North Pacific and the northern flank of ACC (Supplementary Fig. 6c) enhances the amplitude of the SST seasonal cycle. However, these contributions are considerably smaller compared to those resulting from decreased MLD (Fig. 4e).

Another line of evidence favoring the dominance of a shoaling mixed layer is the distribution of amplitude changes across the mixed layer (Fig. 5b). The intensification of the temperature seasonal cycle extends from the surface to the bottom of the mixed layer in all zonal bands. Particularly noteworthy are the changes in the northern subpolar gyres at around 55°N, where the regions correspond closely to areas of the strongest MLD shoaling (Fig. 5b, lines), further confirming the important role of a shallower mixed layer. Another intriguing feature is the amplitude reduction beneath the mixed layer, likely attributed to the overall shallowing of the mixed layer that impedes the propagation of the seasonal cycle signal into deeper ocean layers.

The changes in annual-mean MLD and zonal-mean temperature seasonality detected in climate models are also evident in the Institute of Atmospheric Physics (IAP) (Fig. 5c, d) and Ishii datasets (Supplementary Fig. 8a, b). However, climate models exhibit a deeper-reaching intensification in the northern high latitudes compared to the observations, which is consistent with the deeper climatological MLD in climate models. Notable differences exist in another observational dataset EN4, particularly in the Southern Hemisphere (Supplementary Fig. 8c, d). This discrepancy may be attributed to EN4's use of temperature and salinity relaxation to climatology in areas where data is scarce[36]. Despite this, the strong consistency among results from IAP, Ishii, and model simulations provides further confidence in the robustness of our findings.

**Implications of the intensified SST seasonal cycle**
There is a close relationship between oxygen solubility and SST, wherein the oxygen solubility decreases with rising temperature. Hence, there has been a significant ocean deoxygenation trend since the mid-20th century along with global warming[10,37,38]. However, the impact of global warming on ocean oxygen solubility may extend beyond annual mean trends. With the intensified SST seasonal cycle, the contrast of surface dissolved oxygen between the seasonal maximum in winter and minimum in summer increases (Fig. 6a), demonstrating an intensification at a rate of ~3.7% during 1983–2022 (Fig. 6b). Moreover, models with a more intensified SST seasonal cycle would also see a more intensified seasonal cycle of dissolved oxygen, demonstrating a correlation of 0.84 (Fig. 6c). The spatial patterns of

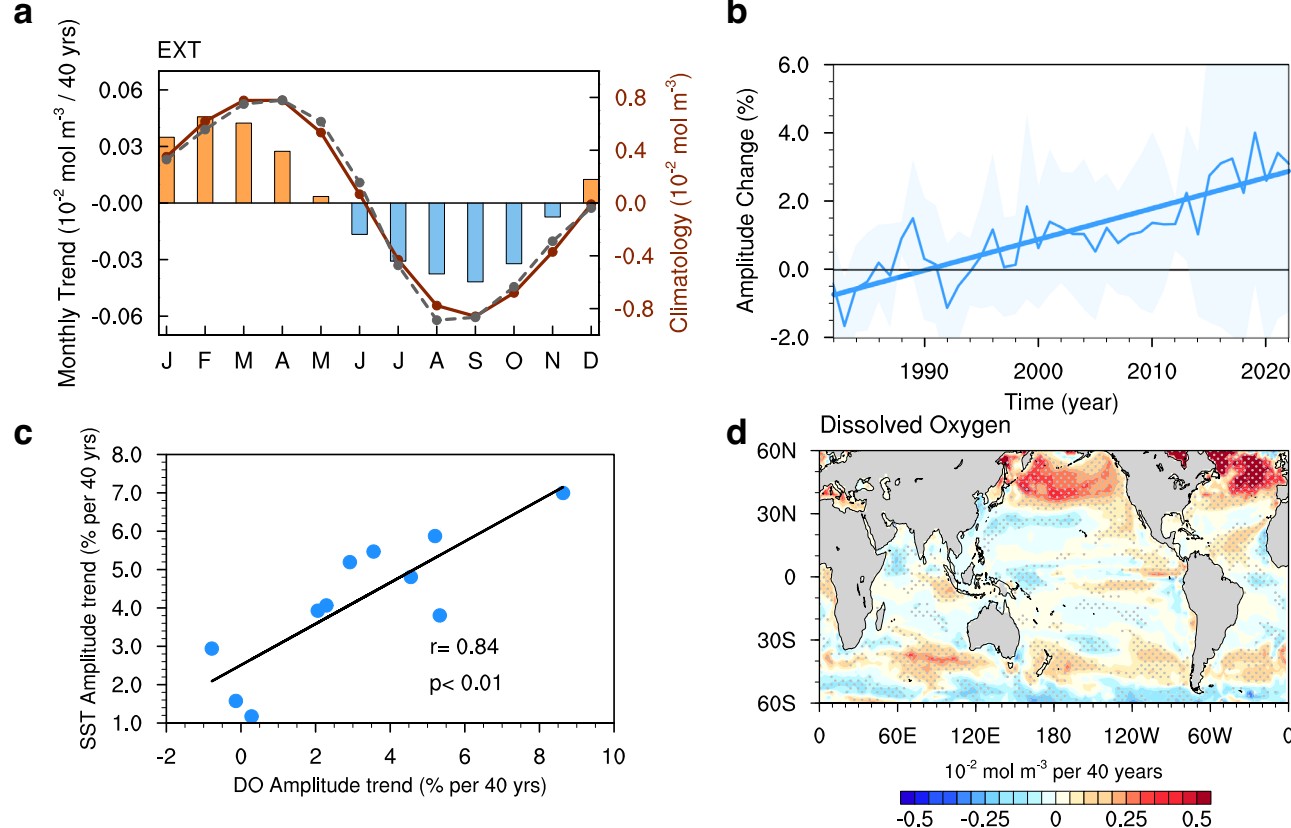

**Fig. 6 | Intensified seasonal cycle of surface dissolved oxygen associated with sea surface temperature (SST) seasonal cycle changes. a** The seasonal cycle of surface dissolved oxygen averaged over the Northern Hemisphere (red dotted line; 0–60 °N) and its trend (bars) during the period 1983–2022 from external forcing simulation ensembles (EXT; mean of CMIP6-MME, ACCESS-ESM1-5 LENS, CanESM5 LENS, and MPI-ESM1-2-LR LENS), and the black dotted line is the mean seasonal cycle over the Northern Hemisphere in World Ocean Atlas 2018. **b** Temporal evolution of changes in the global mean amplitude of surface dissolved oxygen seasonal cycle (unit: %) from EXT relative to the climatological mean of 1983–1992, with standard deviation among four simulation ensembles (pink shading). Linear trend during 1983–2022 are also indicated. **c** An inter-model relationship between the trend of surface dissolved oxygen (DO) seasonal cycle amplitude and the trend of SST seasonal cycle amplitude during 1983–2022. The black line represents the linear regression line. Also shown are the correlation coefficient and corresponding *p*-value. **d** Linear trend of surface dissolved oxygen seasonal cycle amplitude (unit: $10^{-2}$ mol m$^{-3}$ per 40 y) during 1983–2022 from EXT. Stippling indicates where the trend is statistically significant above the 95% confidence level based on Student's *t*-test.

the intensification in these two seasonal cycles (Fig. 1f versus Fig. 6d) are also similar, with a pattern correlation of 0.60. This collectively underscores the impact of the intensification of the SST seasonal cycle on ocean oxygen solubility. Regional analysis reveals a substantial 15.7% enhancement in the seasonal contrast of dissolved oxygen in the subpolar North Atlantic (40–60°N, 60°W-0). Intensifications in the subpolar North Pacific (40–60°N, 140°E-120°W) and Southern Ocean (35–50°S) also reach 5.5% and 6.9%, respectively. These enhancements may foster the occurrence of hypoxic conditions characterized by exceedingly low oxygen concentrations[39] by superimposing upon the long-term oxygen depletion trend.

Apart from dissolved oxygen, the air-sea $CO_2$ flux (FCO2) is significantly influenced by SST variations. Our analysis reveals a notable intensification of FCO2 seasonality during 1983–2022 (Supplementary Fig. 9a). However, its spatial pattern differs from that of intensified SST seasonality. For instance, the intensification in the FCO2 seasonal cycle is notable in subtropical oceans and the southern flank of the ACC, where changes in the SST seasonality are relatively minor or even negative. Conversely, in regions such as the subpolar North Pacific north of 45°N, which experience a remarkable SST seasonal cycle enhancement, the FCO2 seasonal cycle is suppressed. Furthermore, the intermodel correlation between global mean changes in FCO2 and SST seasonal cycles is weak (r = 0.14; Supplementary Fig. 9b), suggesting that mechanisms responsible for

the intensified FCO2 seasonal cycle involve other factors, such as surface winds[40].

## Discussion

Based on observational datasets and large ensemble climate simulations, our study reveals a significant intensification in the seasonal cycle of global SST. Over the period 1983–2022, we observe an increase of approximately 3.9% ± 1.6%, with the strongest intensification occurring in the northern subpolar gyres and the northern flank of the Antarctic Circumpolar Current. The successful replication of these intensifications by a multi-model ensemble of climate simulations, in terms of their temporal evolution and spatial distribution, underscores the crucial role of human-induced external forcings in driving these changes in SST seasonality. The primary driver of this intensification is the increased greenhouse gases, with a discernible impact in the Northern Hemisphere stemming from reduced anthropogenic aerosols since the 1980s. A recent study identified an intensified SST seasonal cycle from 1950 to 2014, which was also mainly attributed to a decreased MLD[41]. In contrast, our findings reveal that this intensified seasonal cycle has emerged even over a shorter time period and extended into the whole mixed layer. Apart from the contribution of decreased MLD, we demonstrate that increased ocean heat uptake and suppressed seasonal amplitude of the surface heat flux also play a role, particularly in the North Pacific and North Atlantic. The intensification

trend in the SST seasonal cycle bears implications for the oceanic[7,42] and terrestrial[43,44] extreme events, alters upper-ocean oxygenation, and compounds existing ecological stressors. In particular, our findings demonstrate that the intensified SST seasonal cycle leads to an intensification in the seasonal cycle of dissolved oxygen, which can impose a variety of stresses on marine organisms, especially in the North Pacific where the "dead zone" is already the most extensive[10]. Therefore, urgent attention and collaborative efforts are essential to confront the challenges posed by the intensifying seasonal cycle of SST.

## Methods

### Observational and reanalysis datasets

In this study, we focus on the amplitude changes of the SST seasonal cycle from 1982 to 2022. We utilize three observational datasets: (1) Extended Reconstructed SST (ERSST) version 5, covering the period from 1854 to the present with a $2° \times 2°$ grid resolution[45], (2) Hadley Centre Sea Ice and SST dataset (HadISST) version 1.1, covering the period from 1870 to the present with a $1° \times 1°$ grid resolution[46], and (3) Optimum Interpolation Sea Surface Temperature (OISST) version 2, covering the period from 1982 to the present with a $0.25° \times 0.25°$ grid resolution[47].

To examine changes in the seasonal cycle of mixed layer temperature and the annual mean MLD from 1982 to 2022, we use three monthly objective analyses ocean dataset: (1) the Institute of Atmospheric Physics (IAP) ocean temperature and salinity analysis, covering the period from 1940 to the present with a $1° \times 1°$ grid resolution and 41 vertical levels for the upper 2,000 m[48], (2) the Ishii v7.3, covering the period from 1955 to 2019 with a $1° \times 1°$ grid resolution and 28 vertical levels for the upper 3000 m[49], and (3) the EN4 data, covering the period from 1900 to present with a $1° \times 1°$ grid resolution and 42 vertical levels extending to beyond 5,000 m[36]. It is important to note that the EN4 dataset applies temperature and salinity relaxation to climatology in regions with sparse data coverage, and thus conducting long-term trend analysis with EN4 could potentially yield problematic results[36]. To ensure consistent analysis, all the above datasets were interpolated onto a $2° \times 2°$ grid using bilinear interpolation.

### Climate model simulations

To estimate the impact of external forcing on the observed intensified SST seasonal cycle during 1982–2022, we use CMIP6 outputs of SST, ocean temperature, zonal velocity, meridional velocity, vertical velocity, surface heat flux, MLD, surface dissolved oxygen, surface $CO_2$ flux from the historical and SSP scenario simulations based on data availability (Supplementary Table 1)[50]. Since CMIP6 historical simulations ended in 2014, we extend them to 2022 using SSP5-8.5 scenarios[51]. Only the first member of each model is used to ensure equal weight in the multi-model ensemble mean analysis. In addition, we use historical and SSP585 SST data from four large ensemble simulations (LENS): (1) 40 ensemble members of the ACCESS-ESM1-5[52], (2) 25 ensemble members of the CanESM5-LE[53], (3) 30 ensemble members of the MPI-ESM1-2-LR[54], and (4) 50 ensemble members of the MIROC6[55]. For the analysis of MLD and surface dissolved oxygen, only the first three LENS are utilized due to data availability.

To assess the contributions of different anthropogenic and natural forcings to the SST seasonal cycle changes, we use SST output from single forcing simulations from the Detection and Attribution Model Intercomparison Project (DAMIP)[56]. These simulations include scenarios with GHGs only, AERs only, StratO3 only, and NATs only simulations. We select seven models that have more than three members for single forcing simulations, and the simulations cover the period of 1982–2020 (Supplementary Table 1). All available members from each model are included in the analysis. All the outputs are interpolated onto a $2° \times 2°$ grid using a bilinear interpolation.

### Amplitude of seasonal cycle

The amplitude of the SST seasonal cycle is determined at each grid point by calculating the difference between the monthly maximum and minimum SST values. Although the timings of maximum and minimum SST can change under climate change, the magnitude of this change is relatively small, typically only a few days. Therefore, we obtain the months of maximum and minimum SST based on the climatology spanning 1970–2000 from the ERSST. These identified months serve as the basis for calculating the amplitude of the seasonal cycle for each year in both observations and simulations. A similar method is applied to surface heat flux, ocean temperature, surface dissolved oxygen, and air-sea $CO_2$ flux data to determine their respective seasonal cycle amplitudes. For ocean temperature and MLD, the climatology is based on the period 1970–2000 from the IAP dataset. For surface heat flux, surface dissolved oxygen and air-sea $CO_2$ flux, the climatology is based on the same period from the CMIP6 multi-model mean.

### Mixed layer depth

CMIP6 models provide the variable *mlotst*, which is the MLD calculated instantaneously on the model timestep. It is defined as the depth at which the potential density exceeds the sea surface density at a criterion of $0.125 \, \text{kg m}^{-3}$. In comparison, our calculations for the IAP dataset utilize monthly potential density with a $0.01 \, \text{kg m}^{-3}$ criterion[57]. The potential density is calculated from temperature and salinity fields using the TEOS-10 Gibbs Seawater (GSW) toolbox. While a commonly used criterion of $0.03 \, \text{kg m}^{-3}$ would yield similar results, the $0.01 \, \text{kg m}^{-3}$ criterion was chosen due to its perfect alignment with the depth at which the intensification in the temperature seasonal cycle can be observed (Fig. 5b, d). Our analysis of MLD based on the IAP, Ishii, and EXT simulations shows similar decreasing trend from 1983 to 2022. However, there is a notable discrepancy observed with the EN4 dataset, possibly due to the fact that EN4 incorporates temperature and salinity relaxation to the climatology in data-sparse regions[36], while IAP and Ishii do not employ such a relaxation method.

### Mixed layer heat budget

We investigated the heat budget of the ocean mixed layer as the following:

$$\frac{\partial T_m}{\partial t} = \frac{Q_{net}}{C_p \rho h} - u_m \cdot \nabla T_m + r, \qquad (1)$$

Here, $T_m$ represents the mixed layer temperature. The thermal forcing term $Q_m$ is determined by the surface heat flux $Q_{net}$ and the mixed layer depth $h$. The constants $C_p$ and $\rho$ are the density and heat capacity of seawater, respectively. $-u_m \cdot \nabla T_m$ represents the horizontal advection term, where $u_m$ is the horizontal velocity averaged over the mixed layer. The vertical entrainment and other unresolved processes are calculated as a residual term $r$. We then evaluated the contributions of different factors to changes in $T_m$ by integrating each tendency term on the right-hand side of the equation over time from January 1982 to December 2022. To obtain the amplitude of the time-integrated tendency terms, we use values corresponding to the months of maximum and minimum $T_m$.

In addition, we decomposed the thermal forcing term into contributions from four factors[14]: annual mean changes in $Q_{net}$, seasonal cycle amplitude changes in $Q_{net}$, annual mean changes in MLD, and seasonal cycle amplitude changes in MLD. To assess the impact of each factor, we recalculated the time-integrated thermal forcing term with one time-evolving factor at a time, while keeping the other three factors constant at their climatological means during the integration. Due to data availability (Supplementary Table 1), the heat budget analysis involves eight models: CAMS-CSM1-0, CAS-ESM2-0, CanESM5, CESM2-WACCM, FGOALS-f3-L, IPSL-CM6A-LR, NESM3.

**Statistical significance test.** The statistical significance of the correlation coefficients is determined by using an "effective sample size" $N^{*}$[58]:

$$N^{*} = N \frac{1 - r_1 r_2}{1 + r_1 r_2},\qquad (2)$$

Where N is the number of available time steps and $r_1$ and $r_2$ are lag-one autocorrelation coefficients of each variable.

## Data availability

The data are available in the following links. ERSST v5 is publicly available at: https://psl.noaa.gov/data/gridded/data.noaa.ersst.v5.html. HadISST v1.1 is publicly available at: https://www.metoffice.gov.uk/hadobs/hadisst/. OISST v2 is publicly available at: https://www.esrl.noaa.gov/psd/data/gridded/data.noaa.oisst.v2.html. IAP is publicly available at http://www.ocean.iap.ac.cn/. Ishii v7.3 is publicly available at https://climate.mri-jma.go.jp/pub/. EN4 is publicly available at http://hadobs.metoffice.com/en4. WOA is publicly available at https://www.nodc.noaa.gov/OC5/woa18/. The CMIP6 and DAMIP data are publicly available at: https://esgf-node.llnl.gov/.

## Code availability

Codes that were used in this study are available from the corresponding authors upon request.

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

## Acknowledgements

We acknowledge the World Climate Research Programme, which, through its Working Group on Coupled Modelling, coordinated and promoted CMIP6. We thank the climate modeling groups for producing and making available their model output, the Earth System Grid Federation (ESGF) for archiving the data and providing access, and the multiple funding agencies who support CMIP6 and ESGF. This work is supported by the National Natural Science Foundation of China (No. 42230405 to F.L. and Y.L.; No. 42175029 to F.S.), the Science and Technology Innovation Project of Laoshan Laboratory (No. LSKJ202202401 to F.L. and Y.L.; No. LSKJ202202201 to F.S.), the Fundamental Research Funds for the Central Universities (No. 202341016 to F.L.; No. 202312006 to F.S.), and the "Youth Innovation Team Program" Team in Colleges and Universities of Shandong Province (No. 2022KJ042 to F.L.). Computing resources are financially supported by Laoshan Laboratory (No. LSKJ202300302).

## Author contributions

F.L. conceived the initial idea, analyzed the data, plotted the figures, and wrote the initial manuscript; F.S. contributed to interpreting and organizing the results and improving the manuscript; Y.L. led the research and improved the manuscript.

## Competing interests

The authors declare no competing interests.
