## [Peer Review File · Nature Communications]

Human-induced intensified seasonal cycle of sea surface temperatureREVIEWER COMMENTS

Reviewer #1 (Remarks to the Author):

General Comments:

This study examined the intensification of SST seasonal cycle using observational data and CMIP6/DAMIP models output. The authors suggested 4.6 (± 1.5)% intensification of SST seasonal cycle's amplitude during 1960-2022, especially, northern subpolar gyre and northern flank of the ACC, where the amplification is up to 10%. CMIP models show similar pattern of SC amplification and employing DAMIP outputs, anthropogenic GHG is more responsible for this reinforcement than aerosol and natural climate variability. Additionally, they show indication of its impacts on ocean biogeochemical properties like dissolved oxygen and ecosystem.

In general, the manuscript is well written and reads quite well. I found this study quite interesting and potential to share more widely with interdisciplinary fields. On the other hand, I have some questions and suggestions for adding more discussions. After the authors consider my points, I would recommend this work published in Nature Communications.

Specific Comments.

1. Figure 3d and f, zonal-mean of MLD from observation and models. Throughout the manuscript, the authors emphasize the MLD shoaling in the higher latitude in NH and SH (subpolar gyre and ACC). I could see some remarkable distribution of shoaling from 2D plots. However, the lines in Figs.3d and f look quite different between observation and models. In the NH (let say, 50-60N), observation (IAP) shows quite a small difference between two periods while model (EXT) shows a clear difference. In the SH, the difference between two periods does not look similar between IAP and EXT. Also, climatological MLD of IAP seems too shallow or very different from EXT (40m and 100m in NH and 60m and 100m). I am wondering how reliable IAP is as a benchmark. And given that IAP is a qualified reference for models, I am curious if the climatological bias of EXT's MLD influences interpretation of results. I would recommend that some argument of this point can be added.

2. While I see a good agreement between SC intensification and MLD shoaling especially, subpolar region. However, I am curious about the mechanism of MLD shoaling and its contribution to SC intensification in those regions. The authors attribute that to the GHG and corresponding ocean heat uptake using DAMIP and referring literatures, but global warming influence dynamics of atmosphere and ocean as well. To me, the authors' argument sounds like one dimensional mechanism. However, in particular, subpolar gyre and ACC are more complicated because of ocean current and sea ice formation (also maybe atmospheric circulation). I am wondering if the changes in those components can also change MLD and seasonal cycle of SST. Actually, the authors mention ACC change due to atmospheric circulation, but how about in the NH, Pacific and Atlantic (for example, Gulf Stream and Kuroshio/Oyashio changes and influencing SST SC and MLD)?

3. Biogeochemical aspect. While the authors show implication for dissolved oxygen and marine ecosystem, I am curious how air-sea carbon dioxide flux would respond to SC intensification. Because subpolar region of NH and SH is a sink of air-sea CO₂ flux, it is important in global carbon cycle. I totally understand there is limitation of space, but really interesting to see any discussion or implication about impacts on air-sea CO₂ flux. The authors would NOT need to add any plots for this point. I would like to see how the authors think about this.

4. Personally, the suppression of SST SC in 1983 and 1992 due to volcanic eruption is quite interesting. I would suppose that this suppression can be attributed to less downwelling shortwave radiation because of abundant aerosol emission. Would the authors see any agreement of MLD deepening and shortwave radiation change in 2D distribution? On the other hand, GHG contribution to intensification of SST SC is due to more re-emission of longwave radiation down to the surface. So, I am wondering if the downwelling longwave radiation at the surface and MLD shoaling match well. Perhaps, this might strengthen the authors' discussion on causality of SST SC intensification due to MLD shoaling.

Minor Comments.

1. Fig1a and b. The caption is not clear.
2. Line 133. Blue => light blue?
3. Fig.2a. The thickness of line in the legend can be widen a bit more?
4. Line 136. Europe, North America, and China?
5. Lines 149-150. But, not significant.

Reviewer #2 (Remarks to the Author):

Review of Human-induced intensified seasonal cycle of sea surface temperature by Liu et al.

In this study, the authors use several observational data products and CMIP6 model results to investigate the intensification of the SST seasonal cycle from 1960 to the present. The results show that there has been a 4.6% increase in the SST seasonal cycle in the global ocean, with the magnitude of the increase varying widely among ocean regions. Using climate model output from DAMIP, they conclude that most of the increase in SST seasonal cycle amplitude has been driven by anthropogenic greenhouse gas emissions. The authors argue that the increase in SST seasonal cycle amplitude is mainly due to a decrease in the annual mean MLD. The authors also suggest that the increase in the SST seasonal cycle may have affected the seasonal cycle of sea surface dissolved oxygen through changes in its surface solubility due to the amplification of the SST seasonal cycle.

As the authors mentioned in the introduction, there have already been many studies that have proposed the concept of long-term changes in the SST seasonal cycle itself using observational data, as well as

studies that have used climate models to predict future changes in the amplitude of the SST seasonal cycle. Therefore, the novelty of this study lies only in the spatial coverage of the assessment of historical long-term trends. The method used for attributing changes in the SST seasonal cycle is an established one (using existing outputs from DAMIP). The authors' conclusion on the mechanism is also not a new hypothesis, as it has been proposed in previous studies on the prediction of future changes in SST seasonal cycle amplitudes using climate models. The implications for changes in the seasonal amplitude of sea surface dissolved oxygen may be a new aspect of this study, but the validity of the model dissolved oxygen data used has not been fully investigated.

Throughout the manuscript, I have three main concerns and several more to follow.

Major concerns:

(1) Reliability of SST data before 1982

As the authors know, the number of SST observations has changed dramatically since the advent of the satellite observations in 1982, i.e., the number of pre-1982 observations is remarkably small. It is a well-known fact that the number of observations is fatally low, especially in the Southern Ocean and in the deep winter season in high latitudes, and that there are not enough observations to fully resolve the SST seasonal cycle (see, for example, the spatial coverage and number of observations archived in the World Ocean Database). I have doubts about the inclusion of the pre-1982 period in the analysis of this study based on some results read from the figures in the manuscript, such as

-Figure S1c-f: The winter trends in each hemisphere, which are important for defining the amplitudes of the seasonal cycle, differ between products.

-Figure S3: The magnitude and sign of the trend in the Southern Ocean varies greatly when the time period is changed (1960- or 1982-) for both ERSST and HadISST (as also the authors mention in L103 but assume as minor). Note that the consistency between the two products does not resolve this concern, as the two products use similar data sources prior to 1982, just with different mapping methods.

-Figure S1 a and b: There appears to be a large discrepancy between the model and observations in Figure S1 a and b (for a fair comparison, please use the same y-axis range for these panels). This seems to be due to the strong questionable trend in the Indian Ocean sector of the Southern Ocean that appears in the calculation since 1960. It is true that there could be a bias in the model, but if the observations contain errors, wouldn't it be better to start from 1982 so that the model and observations are more consistent?

-Figure 1c: There seems to be no significant trend before 1980.

(2) Conclusion about the mechanism

The authors concluded that the decrease in the annual mean MLD is the main "mechanism" for the increase in the amplitude of the SST seasonal cycle by showing only an intermodel "correlation" between changes in the annual mean MLD and SST amplification. However, this is only for the global mean trend, and the spatial correspondence of the signals is not very good, although the authors say it is good (comparing Fig. 3c to 1e). Even in the zonal mean (Fig. 3d), there is no dominant mixed layer thinning north of 50N, where the seasonal amplification is greatest, and the seasonal SST amplitude is not enhanced despite a strong mixed layer shallowing trend at 30S. Again, this may be due to errors in the data product (in both IAP and Ishii) due to the lack of vertical profiles prior to the Argo era, as the model appears to be better (Fig. 3f).

Since there may be several other mechanisms for the seasonal amplification of the SST besides the one rejected by the authors (changes in the seasonal amplitude of the sea surface flux) (e.g. changes in the annual mean sea surface heat flux and changes in the seasonal mean MLD), I believe that the authors can only point out the relationship between the annual mean MLD and the SST amplification, and not determine the mechanism (as the authors also mention in L203).

3) Possibilities for modeled dissolved oxygen

In general, even the Earth system models (ESMs) in the CMIP6 generation suffer from serious dissolved oxygen biases mainly due to incomplete parameterizations of air-sea exchange process and biological processes. Although it is difficult to obtain the observational trend of monthly sea surface dissolved oxygen due to lack of historical observational data, the model climatological seasonal cycle should be validated using recent climatological observational data products, for example, World Ocean Atlas (<https://www.ncei.noaa.gov/products/world-ocean-atlas>) and Gridded Ocean Biogeochemistry from Artificial Intelligence – Oxygen (<https://www.ncei.noaa.gov/access/metadata/landing-page/bin/iso?id=gov.noaa.nodc:0259304>).

Other important concerns:

L34: In references 8-11, I could not find any reference to “the effect of changes in the SST seasonal cycle on marine ecosystems”. Please elaborate this sentence.

L82 : What does the plus/minus range mean? Please clarify the statistical methodology in the text.

L93: This is true only for the case of the Northern hemisphere (Fig. 1a and b). It seems that there are some differences in the case of the Southern hemisphere (Fig. S3a and b)

L113: How did the authors estimate the degree of freedom for the statistical test? Using the number of data grid point as the degree of freedom is overestimate because adjacent grids are not necessarily independent. Please clarify the statistical method.

L116 “Some discrepancies between observations and models may arise either from the influence of low-frequency internal variability”: Could you estimate the range of internal variability (ensemble standard deviation) using the ensemble members and compare the range with the discrepancies between observations and models?

L215 “Another intriguing feature is the widespread amplitude reduction beneath the mixed layer, attributed to the overall shallowing of the mixed layer that impedes the propagation of the seasonal cycle signal into deeper ocean layers.”: Interesting.

L242: “deoxygenation” rather than “oxygenation”?

L243 “The enhanced seasonal contrast of upper ocean oxygen level may foster the occurrence of hypoxic conditions characterized by exceedingly low oxygen concentrations by superimposing upon the long-term oxygen depletion trend over the past 60 years”: To quantify this impact, please show the spatial map of the relative changes in the sea surface oxygen amplitude (i.e., oxygen amplitude changes over

the past 60 years divided by climatological seasonal amplitude).

L325 : "This calculation is based on the climatology of the period 1970-2000 from the ERSST": I could not get this. Please elaborate this.

L338: "The MLD derived with the 0.01 kg m⁻³ threshold aligns perfectly with regions that exhibit an intensified seasonal cycle of upper ocean temperature at all latitudes": I think this is result and thus this is not placed in the "Method".

Figure 1a and b: The left y-axis should be "Monthly trend" rather than "Amplitude trend"?

Figure S6 and Method: Why did not the authors use observational heat flux product (OAFlux <https://climatedataguide.ucar.edu/climate-data/oaflux-objectively-analyzed-air-sea-fluxes-global-oceans>, and JOFURO3 <https://www.j-ofuro.com/en/>)? Observational data is of limited duration, but should be used for validation of atmospheric reanalysis.

Figure S6b: This analysis also should be done for each hemisphere like Fig. S7.

L176: What about the spatial correspondence between the heat flux changes and SST amplitude change?

Data: Is there any reason why the authors did not use the CESM2 large ensemble (<https://esd.copernicus.org/articles/12/1393/2021/>)? The CESM2 is one of CMIP6 generation models and it has outputs from single forcing experiment (<https://journals.ametsoc.org/view/journals/clim/aop/JCLI-D-22-0666.1/JCLI-D-22-0666.1.xml>).

Method: 0.01 kg m⁻³ is too small for the threshold when determining the MLD from the Grid data product. As noted in the reference, 0.01 kg m⁻³ is a value to use for raw CTD profiles, rather 0.1 kg m⁻³ or 0.125 kg m⁻³ is more common because vertical profiles in gridded product are vertically smoothed.

Figure S4a: It would be better to show the spatial map of the CMIP6 model spread (standard deviation) next to Fig. S4a.

We thank the reviewers for their insightful comments and detailed suggestions, which help improve the manuscript. We have focused on satellite-era data (after 1982) to ensure the robustness of our results. In addition, we have conducted a comprehensive heat budget analysis to demonstrate the primary role of decreased MLD in intensifying the SST seasonal cycle. We appreciate the valuable input from both reviewers, which has motivated us to improve the quality of our work. In the following, we provide a point-by-point response to both reviewers.

Note that major changes are highlighted in red in the revised text.

Reviewer #1 (Remarks to the Author):

General Comments:

This study examined the intensification of SST seasonal cycle using observational data and CMIP6/DAMIP models output. The authors suggested 4.6 (± 1.5)% intensification of SST seasonal cycle's amplitude during 1960-2022, especially, northern subpolar gyre and northern flank of the ACC, where the amplification is up to 10%. CMIP models show similar pattern of SC amplification and employing DAMIP outputs, anthropogenic GHG is more responsible for this reinforcement than aerosol and natural climate variability. Additionally, they show indication of its impacts on ocean biogeochemical properties like dissolved oxygen and ecosystem.

In general, the manuscript is well written and reads quite well. I found this study quite interesting and potential to share more widely with interdisciplinary fields. On the other hand, I have some questions and suggestions for adding more discussions. After the authors consider my points, I would recommend this work published in Nature Communications.

Response: Thank you for your positive feedback and valuable suggestions. We have made substantial revisions to improve the manuscript. Specifically, we have performed a quantification analysis using a mixed-layer heat budget to assess the relative contributions of various factors, which provides compelling evidence that decreased MLD is the primary driver of SST seasonality intensification. In addition, we have expanded our discussion to include an examination of the discrepancies between modeled and simulated MLD, and the implication of the intensified seasonal amplitude for surface CO₂ flux.

Below are our point-by-point responses to your comments.

Specific Comments.

1. Figure 3d and f, zonal-mean of MLD from observation and models. Throughout the manuscript, the authors emphasize the MLD shoaling in the higher latitude in NH and SH (subpolar gyre and ACC). I could see some remarkable distribution of shoaling from 2D plots. However, the lines in Figs.3d and f look quite different between observation and models. In the NH (let say, 50-60N), observation (IAP) shows quite a small difference between two periods while model (EXT) shows

a clear difference. In the SH, the difference between two periods does not look similar between IAP and EXT. Also, climatological MLD of IAP seems too shallow or very different from EXT (40m and 100m in NH and 60m and 100m). I am wondering how reliable IAP is as a benchmark. And given that IAP is a qualified reference for models, I am curious if the climatological bias of EXT's MLD influences interpretation of results.

Response: Thank you for bringing this to our attention. We have revised the analysis to present trends in zonal-mean MLD over the period 1983-2022 (purple line in Fig. R1a), rather than the previous average over 2013-2022. This adjustment helps to mitigate the influence of internal variability on the results and provides a more accurate representation of long-term trends. As a result, the discrepancy in MLD between the IAP observations and CMIP6 simulations in the 50-60N region has been significantly reduced, with both showing a notable decrease in MLD.

However, the difference between observations and simulations in the SH is still large. One possible reason is the less reliable observational datasets in the SH. In addition, differences in the criteria used to calculate MLD between the IAP dataset and CMIP6 simulations may contribute to discrepancies. While CMIP6 calculates MLD *instantaneously* using a criterion of 0.125 kg/m³, our calculations for the IAP dataset utilize *monthly* temperature and salinity fields with a criterion of 0.01 kg/m³. This 0.01 kg/m³ criterion was chosen because the monthly fields are smoother compared to the CMIP6 instantaneous fields, and the MLD defined in this way aligns well with the intensified temperature seasonal cycle (blue line in Fig. R1b). If we were to use the 0.125 kg/m³ criterion to define MLD in the IAP dataset, the results would indicate excessive depth, especially at high latitudes (purple line in Fig. R1b), and would deviate significantly from the intensification trend.

We have incorporated this discussion on the potential causes for the discrepancy between observed and simulated MLD into the Method Section for clarity (Lines 354-361).

Fig. R1| a, Linear trend of the zonal mean seasonal cycle of temperature (unit: m per 40yr) during 1983-2022 from the IAP dataset. The blue denotes the annual mean MLD averaged over 1983-1992, calculated with 0.01 kg m⁻³ threshold, and the red line denotes its response over 1983-

2022. **b**, same as a, but the blue and red lines denote the annual mean MLD averaged over 1983-1992 calculated with 0.01 and 0.125 kg m⁻³ threshold, respectively.

2. While I see a good agreement between SC intensification and MLD shoaling especially, subpolar region. However, I am curious about the mechanism of MLD shoaling and its contribution to SC intensification in those regions. The authors attribute that to the GHG and corresponding ocean heat uptake using DAMIP and referring literatures, but global warming influence dynamics of atmosphere and ocean as well. To me, the authors' argument sounds like one dimensional mechanism. However, in particular, subpolar gyre and ACC are more complicated because of ocean current and sea ice formation (also maybe atmospheric circulation). I am wondering if the changes in those components can also change MLD and seasonal cycle of SST. Actually, the authors mention ACC change due to atmospheric circulation, but how about in the NH, Pacific and Atlantic (for example, Gulf Stream and Kuroshio/Oyashio changes and influencing SST SC and MLD)?

Response: We appreciate your perspective comment. To provide a comprehensive analysis of the underlying mechanisms of the intensified SST seasonal cycle, we have conducted a mixed layer heat budget analysis. This approach allows us to quantitatively assess the contributions of various factors, including the *horizontal advection*, the thermal forcing term, (influenced by net surface heat flux and MLD), and other residual processes (see Methods for more details).

Our analysis reveals a limited role of ocean circulation changes in shaping the SST seasonality changes, both in its temporal evolution (Fig. R2a, blue line) and trend pattern (Fig. R2c). In addition, the thermal forcing term is found to be dominant in driving the intensified SST seasonal cycle (Fig. R2a,b).

Furthermore, we have decomposed the thermal forcing term into contributions from different factors, including annual mean changes in SHF, seasonal cycle amplitude changes in SHF, annual mean changes in MLD, and seasonal cycle amplitude changes in MLD. This analysis reveals that the decrease in MLD is the primary contributor (Fig. 4e), driving the overall intensification of the seasonal cycle.

In the revised manuscript, the section “**Mechanisms of the intensified SST seasonal cycle**” has been rewritten to include these findings (Lines 175-222), providing a compelling line of evidence for the dominant role of decreased MLD.

Fig. R2 | Causes of intensification in SST seasonal cycle. **a**, Temporal evolution of changes in the global mean amplitude of mixed layer temperature (Tm; black line) seasonal cycle (unit: %) from CMIP6 MME relative to the mean of 1983-1992, and its decomposition into contributions from the thermal forcing term (Qm; red line), the horizontal advection term (HADV; blue line), and the residual term (R; orange line). **b-d**, Linear trend of SST seasonal cycle amplitude during 1983-2022 (unit: °C per 40yr) due to (b) the thermal forcing term, (c) the horizontal advection term, and (d) the residual term from CMIP6 MME.

3. Biogeochemical aspect. While the authors show implication for dissolved oxygen and marine ecosystem, I am curious how air-sea carbon dioxide flux would respond to SC intensification. Because subpolar region of NH and SH is a sink of air-sea CO₂ flux, it is important in global carbon cycle. I totally understand there is limitation of space, but really interesting to see any discussion or implication about impacts on air-sea CO₂ flux. The authors would NOT need to add any plots for this point. I would like to see how the authors think about this.

Response: Thank you for your insightful comment. We have explored the response of air-sea CO₂ flux (FCO₂) seasonal amplitude using the CMIP6 field *f_{gco2}* during the period 1982-2022.

Our analysis reveals a significant intensification of FCO₂ seasonality (Fig. R3a). However, its spatial pattern differs from that of the intensified SST seasonality. For instance, the intensification of the FCO₂ seasonal cycle is notable in the subtropical oceans and the southern flank of the ACC,

where changes in the SST seasonality are relatively small or even negative. Conversely, the FCO2 seasonal cycle is suppressed in regions such as the subpolar North Pacific north of 45°N, which experience a remarkable enhancement of the SST seasonal cycle. Furthermore, the intermodel correlation between global mean changes in FCO2 and SST seasonal cycles is low ($r=0.1$; Fig. R3b), indicating that the mechanisms responsible for the intensified FCO2 seasonal cycle involve other factors, such as surface winds and biological processes (Lerner et al. 2021).

The above discussion has been incorporated into the revised manuscript (Line 266-276& Supplementary Fig. 9) to provide a more comprehensive analysis of the biogeochemical implications of the intensified SST seasonal cycle.

Fig. R3 | Intensified seasonal cycle of surface CO2 flux unrelated to changes in SST seasonal cycle. **a**, Linear trend of surface CO2 flux seasonal cycle amplitude (unit: $\text{g C m}^{-2} \text{ year}^{-1}$ per 40yr) during 1983-2022 from CMIP6. **b**, an inter-model relationship between the trend of surface dissolved oxygen seasonal cycle amplitude and the trend of SST seasonal cycle amplitude during 1983-2022.

References:

Lerner, P., A. Romanou, M. Kelley, J. Romanski, R. Ruedy, and G. Russell, 2021: Drivers of Air-Sea CO2 Flux Seasonality and its Long-Term Changes in the NASA-GISS Model CMIP6 Submission. *Journal of Advances in Modeling Earth Systems*, **13**, 1–33, <https://doi.org/10.1029/2019MS002028>.

4. Personally, the suppression of SST SC in 1983 and 1992 due to volcanic eruption is quite interesting. I would suppose that this suppression can be attributed to less downwelling shortwave radiation because of abundant aerosol emission. Would the authors see any agreement of MLD deepening and shortwave radiation change in 2D distribution? On the other hand, GHG contribution to intensification of SST SC is due to more re-emission of longwave radiation down to the surface. So, I am wondering if the downwelling longwave radiation at the surface and MLD

shoaling match well. Perhaps, this might strengthen the authors' discussion on causality of SST SC intensification due to MLD shoaling.

Response: Thank you for raising this point. As you pointed out, changes in surface heat flux (SHF) and MLD, as well as changes in other ocean dynamical processes, may all contribute to changes in the SST seasonal cycle. To address this, we have conducted a comprehensive heat budget analysis to quantitatively assess the contributions of different factors, as detailed in our response to your major comment #2. Our analysis reveals distinct mechanisms underlying the suppression of the SST seasonal cycle during the 1983 and 1992 volcanic events.

Specifically, we found that the suppression in 1992 is primarily induced by the thermal forcing effect alone, whereas the suppression in 1983 is influenced by both thermal forcing and ocean circulation changes (Fig. R4a). Moreover, the controlling factors for the thermal forcing term in the two events are also different: the 1992 event is mainly associated with changes in the MLD (Fig. R4c), while the 1983 event is largely contributed by amplitude changes in the SHF (Fig. R4b). Given our focus on the long-term trend in SST seasonal cycle changes, we have deferred detailed investigations of these events to future analyses.

Regarding the relationship between GHG-induced heat uptake and SST seasonal cycle intensification, our analysis indicates that although the ocean has absorbed more heat over the past four decades, its direct impact on SST seasonal cycle intensification is limited (Fig. 4c, e). Instead, we found that the decreased MLD plays a significant role in driving the intensification (Fig. 4a).

In the revised manuscript, we have thoroughly discussed the contribution of different factors to the long-term intensification trend based on the mixed layer budget analysis, as outlined in Lines 182-222.

Fig. R4 | Mechanisms for the reduced SST seasonal cycle in 1983 and 1992. **a**, Detrended time series of global mean amplitude of mixed layer temperature (Tm; black line) seasonal cycle (unit: °C) from CMIP6 MME, and its decomposition into contributions from the thermal forcing term (Qm; red line), the horizontal advection term (HADV; blue line), and the residual term (R; orange line). **b**, Anomalies of the SST seasonal amplitude due to thermal forcing (Qm) at year 1983, as well as its decomposition into contributions from changes in annual mean MLD (MLD ANN), changes in seasonal amplitude of MLD (MLD AMP), changes in annual mean SHF (SHF ANN), and changes in seasonal amplitude of SHF (SHF AMP). **c**, Same as b, but for year 1992.

Minor Comments.

1. Fig1a and b. The caption is not clear.

Response: Thank you, and we have updated the caption to clarify that the dotted lines represent the SST seasonal cycle climatology, while the bars represent the trends.

2. Line 133. Blue => light blue?

Response: Thank you, and it has been corrected.

3. Fig.2a. The thickness of line in the legend can be widen a bit more?

Response: Done.

4. Line 136. Europe, North America, and China?

Response: Revised as suggested.

5. Lines 149-150. But, not significant.

Response: Thank you for pointing this out, and we have removed the description in the revision.

Reviewer #2 (Remarks to the Author):

Review of Human-induced intensified seasonal cycle of sea surface temperature by Liu et al.

In this study, the authors use several observational data products and CMIP6 model results to investigate the intensification of the SST seasonal cycle from 1960 to the present. The results show that there has been a 4.6% increase in the SST seasonal cycle in the global ocean, with the magnitude of the increase varying widely among ocean regions. Using climate model output from DAMIP, they conclude that most of the increase in SST seasonal cycle amplitude has been driven by anthropogenic greenhouse gas emissions. The authors argue that the increase in SST seasonal cycle amplitude is mainly due to a decrease in the annual mean MLD. The authors also suggest that the increase in the SST seasonal cycle may have affected the seasonal cycle of sea surface dissolved oxygen through changes in its surface solubility due to the amplification of the SST seasonal cycle.

As the authors mentioned in the introduction, there have already been many studies that have proposed the concept of long-term changes in the SST seasonal cycle itself using observational data, as well as studies that have used climate models to predict future changes in the amplitude of the SST seasonal cycle. Therefore, the novelty of this study lies only in the spatial coverage of the assessment of historical long-term trends. The method used for attributing changes in the SST seasonal cycle is an established one (using existing outputs from DAMIP). The authors' conclusion on the mechanism is also not a new hypothesis, as it has been proposed in previous studies on the prediction of future changes in SST seasonal cycle amplitudes using climate models. The implications for changes in the seasonal amplitude of sea surface dissolved oxygen may be a new aspect of this study, but the validity of the model dissolved oxygen data used has not been fully investigated.

Throughout the manuscript, I have three main concerns and several more to follow.

Response: We appreciate your perceptive comments, which have helped us improve the manuscript substantially. As you can see from the revised version of our manuscript, considerable efforts have been made to address your comments, particularly in the following areas:

- (1) **Use of data after 1982:** To ensure the robustness of our results, we have focused on satellite-era data.
- (2) **Quantification analysis based on a heat budget:** We have conducted a mixed-layer heat budget analysis to quantify the contribution of different factors to the intensification of the SST seasonal cycle. Through this analysis, we have identified decreased MLD as the primary driver of SST seasonality intensification.
- (3) **Validation of dissolved oxygen dataset:** We have validated model simulations against the WOA18 to ensure reliability in simulating the climatological seasonal cycle of surface dissolved oxygen.

- (4) **Internal variability:** To assess the influence of internal variability on our results, we have estimated the model spread and found that most of the discrepancy falls within the range of internal variability.

Below are our point-by-point responses to your comments.

Major concerns:

(1) Reliability of SST data before 1982

As the authors know, the number of SST observations has changed dramatically since the advent of the satellite observations in 1982, i.e., the number of pre-1982 observations is remarkably small. It is a well-known fact that the number of observations is fatally low, especially in the Southern Ocean and in the deep winter season in high latitudes, and that there are not enough observations to fully resolve the SST seasonal cycle (see, for example, the spatial coverage and number of observations archived in the World Ocean Database). I have doubts about the inclusion of the pre-1982 period in the analysis of this study based on some results read from the figures in the manuscript, such as

-Figure S1c-f: The winter trends in each hemisphere, which are important for defining the amplitudes of the seasonal cycle, differ between products.

-Figure S3: The magnitude and sign of the trend in the Southern Ocean varies greatly when the time period is changed (1960- or 1982-) for both ERSST and HadISST (as also the authors mention in L103 but assume as minor). Note that the consistency between the two products does not resolve this concern, as the two products use similar data sources prior to 1982, just with different mapping methods.

-Figure S1 a and b: There appears to be a large discrepancy between the model and observations in Figure S1 a and b (for a fair comparison, please use the same y-axis range for these panels). This seems to be due to the strong questionable trend in the Indian Ocean sector of the Southern Ocean that appears in the calculation since 1960. It is true that there could be a bias in the model, but if the observations contain errors, wouldn't it be better to start from 1982 so that the model and observations are more consistent?

-Figure 1c: There seems to be no significant trend before 1980.

Response: Thank you for raising this issue. We acknowledge the limitations associated with pre-satellite era SST observations, particularly in regions such as the Southern Ocean and during the winter season at high latitudes. Taking your suggestion, we have decided to focus our analysis on satellite-era data from 1982 to 2022 in the revised manuscript.

While there are differences in the trend patterns between observations and simulations, we are pleased to note that our main findings regarding the intensification of SST seasonality in the northern subpolar gyres and the northern flank of the ACC remain consistent across all datasets.

We thank you for highlighting this important issue, which helped to improve the robustness of our results.

(2) Conclusion about the mechanism

The authors concluded that the decrease in the annual mean MLD is the main "mechanism" for the increase in the amplitude of the SST seasonal cycle by showing only an intermodel "correlation" between changes in the annual mean MLD and SST amplification. However, this is only for the global mean trend, and the spatial correspondence of the signals is not very good, although the authors say it is good (comparing Fig. 3c to 1e). Even in the zonal mean (Fig. 3d), there is no dominant mixed layer thinning north of 50N, where the seasonal amplification is greatest, and the seasonal SST amplitude is not enhanced despite a strong mixed layer shallowing trend at 30S. Again, this may be due to errors in the data product (in both IAP and Ishii) due to the lack of vertical profiles prior to the Argo era, as the model appears to be better (Fig. 3f).

Since there may be several other mechanisms for the seasonal amplification of the SST besides the one rejected by the authors (changes in the seasonal amplitude of the sea surface flux) (e.g. changes in the annual mean sea surface heat flux and changes in the seasonal mean MLD), I believe that the authors can only point out the relationship between the annual mean MLD and the SST amplification, and not determine the mechanism (as the authors also mention in L203).

Response: We appreciate the insightful comments. We have made significant revisions to address the concerns raised about the conclusion regarding the underlying mechanisms.

First, we would like to address the difference between observed and simulated zonal mean MLD in the 50°-60°N region. This difference has been substantially reduced after presenting the trends in zonal-mean MLD over the period 1983-2022 (Fig. 5d, purple line). The previous analysis, based on the average over 2013-2022, may have been strongly influenced by decadal variability.

Recognizing the limitations of observational datasets, particularly in regions such as the Southern Ocean, we have primarily relied on CMIP6 simulations in the revised manuscript, as these models can capture the overall magnitude and spatial pattern of the intensification trend in key regions of interest (Fig. 1), and the need to perform a detailed heat budget analysis.

To delve deeper into the underlying mechanisms, we conducted a careful mixed layer heat budget analysis instead of the correlation analysis in the previous version. This approach allows us to quantitatively assess the contributions of various factors, including the thermal forcing term Q_m influenced by the net surface heat flux and MLD, horizontal advection, and other residual processes (see Methods for more details). Through this approach, we identified decreasing MLD as the primary driver over most of the global ocean during the period 1982-2022.

Specifically, during 1982-2022, Q_m exhibits positive trends over most of the global ocean (Fig. R5b), particularly in the northern subpolar gyres. The residual term emerges as the major contributor to the seasonal intensification in the equatorial oceans and also plays a role in the intensification on the northern flank of the ACC (Fig. R5d). In contrast, the contribution of horizontal advection is considered negligible (Fig. R5c).

Further investigation is warranted into the primary factors shaping Q_m . These factors include decreased MLD (Fig. R6a), reduced seasonal amplitude of MLD (Fig. R6b), increased heat uptake in the surface ocean (Fig. R6c), and enhanced amplitude of SHF over most of the global ocean (Fig. R6d). Through detailed decomposition, our analysis highlights the predominant role of the shallower MLD (Fig. R6e). Changes in the annual mean and seasonal amplitude of SHF also play a role in the regional intensification of the SST seasonal cycle (Fig. R6c, d).

In the revision, we have rewritten the section “**Mechanisms of the intensified SST seasonal cycle**” to incorporate these important findings (Lines 175-222), which provide a compelling line of evidence for the dominant role of the decreasing MLD. Thank you for your valuable suggestions.

Fig. R5 | Causes of intensification in SST seasonal cycle. a, Temporal evolution of changes in the global mean amplitude of mixed layer temperature (T_m ; black line) seasonal cycle (unit: %) from CMIP6 MME relative to the mean of 1983-1992, and its decomposition into contributions from the thermal forcing term (Q_m ; red line), the horizontal advection term (HADV; blue line),

and the residual term (R; orange line). **b-d**, Linear trend of SST seasonal cycle amplitude during 1983-2022 (unit: °C per 40yr) due to (b) the thermal forcing term, (c) the horizontal advection term, and (d) the residual term from CMIP6 MME.

Fig. R6 | Dominance of shoaling mixed layer in amplified SST seasonal cycle. **a**, Linear trends of the SST amplitude during 1983-2022 (unit: °C per 40yr) due to contributions from changes in (a) annual mean MLD (MLD ANN), (b) changes in seasonal amplitude of MLD (MLD ANN), (c) changes in annual mean SHF (SHF ANN), and (d) changes in seasonal amplitude of SHF (SHF AMP). **e**, Linear trends of the global-mean SST amplitude due to thermal forcing (Qm; gray bar), as well as its decomposition into different contributing factors (red bars).

3) Possibilities for modeled dissolved oxygen

In general, even the Earth system models (ESMs) in the CMIP6 generation suffer from serious dissolved oxygen biases mainly due to incomplete parameterizations of air-sea exchange process and biological processes. Although it is difficult to obtain the observational trend of monthly sea surface dissolved oxygen due to lack of historical observational data, the model climatological seasonal cycle should be validated using recent climatological observational data products, for example, World Ocean Atlas (<https://www.ncei.noaa.gov/products/world-ocean-atlas>) and Gridded Ocean Biogeochemistry from Artificial Intelligence – Oxygen (<https://www.ncei.noaa.gov/access/metadata/landing-page/bin/iso?id=gov.noaa.nodc:0259304>).

Response: Thank you for your comment. We have conducted a comparison of the climatological amplitude of the dissolved oxygen seasonal cycle in the WOA18 and CMIP6. Our analysis, as shown in Fig. R7, demonstrates generally good agreement in terms of the spatial patterns and magnitudes.

In the revised manuscript, we have updated Fig. 5a to include the seasonal cycle of dissolved oxygen averaged over the Northern Hemisphere, represented by the dotted gray line.

Fig. R7 | Consistency between observed and simulated seasonal cycle of dissolved oxygen. a, b, climatological seasonal cycle amplitude of surface ocean dissolved oxygen from (a) WOA18 and (b) EXT.

Other important concerns:

L34: In references 8-11, I could not find any reference to “the effect of changes in the SST seasonal cycle on marine ecosystems”. Please elaborate this sentence.

Response: The following literature has been cited:

Mueter, F. J., C. Bross, K. F. Drinkwater, K. D. Friedland, J. A. Hare, G. L. Hunt, W. Melle, and M. Taylor, 2009: Ecosystem responses to recent oceanographic variability in high-latitude Northern Hemisphere ecosystems. *Progress in Oceanography*, **81**, 93–110, <https://doi.org/10.1016/j.pocean.2009.04.018>.

Ottersen, G., S. Kim, G. Huse, J. J. Polovina, and N. C. Stenseth, 2010: Major pathways by which climate may force marine fish populations. *Journal of Marine Systems*, **79**, 343–360, <https://doi.org/10.1016/j.jmarsys.2008.12.013>.

L82: What does the plus/minus range mean? Please clarify the statistical methodology in the text.

Response: It represents the 5-95% confidence level of the linear trend, indicating the range within which 90% of the data falls. We have made it clear in the revised text (Line 82).

L93: This is true only for the case of the Northern hemisphere (Fig. 1a and b). It seems that there are some differences in the case of the Southern hemisphere (Fig. S3a and b)

Response: Thank you for your comment. We have noted the discrepancy between the observed and simulated amplitude changes in the SH, particularly in the western South Pacific and the Southern Ocean around 50°S. The lower pattern correlation in the SH (0.24 compared to 0.72 in the NH) suggests the influence of large observational uncertainties, particularly in the data-sparse Southern Ocean. Other factors, such as the low-frequency internal variability within the climate system and model deficiencies, may also have contributed to these differences. The difference between OBS and EXT is smaller than the value of the ensemble standard deviation over the majority of global oceans (Fig. R8f).

We have highlighted this interhemispheric contrast in the revised text (Lines 120-124). Thank you for pointing out this important issue.

L113: How did the authors estimate the degree of freedom for the statistical test? Using the number of data grid point as the degree of freedom is overestimate because adjacent grids are not necessarily independent. Please clarify the statistical method.

Response: Thank you. An “effective sample size” N^* (Bretherton et al. 1999) is calculated and used to estimate the degrees of freedom in the significance test of correlation:

$$N^* = N \frac{1-r_1 r_2}{1+r_1 r_2},$$

where N is the number of available time steps and r_1 and r_2 are lag-one autocorrelation coefficients of each variable.

We have added a statement on the statistical significance test in the Methods section (Lines 383-387).

Reference:

Bretherton, C. S., M. Widmann, V. P. Dymnikov, J. M. Wallace, and I. Bladé, 1999: The effective number of spatial degrees of freedom of a time-varying field. *Journal of Climate*, **12**, 1990–2009, [https://doi.org/10.1175/1520-0442\(1999\)012<1990:TENOSD>2.0.CO;2](https://doi.org/10.1175/1520-0442(1999)012<1990:TENOSD>2.0.CO;2).

L116 “Some discrepancies between observations and models may arise either from the influence of low-frequency internal variability”: Could you estimate the range of internal variability (ensemble standard deviation) using the ensemble members and compare the range with the discrepancies between observations and models?

Response: Thank you for your suggestion. In response, we have estimated the internal variability as the ensemble standard deviation of CMIP6 and other LENS ensemble members, respectively (Fig. R8). All the ensembles exhibit significant internal variability in regions where the seasonal intensification is prominent, particularly in the subpolar North Atlantic and Pacific.

We found that the difference between OBS and EXT is smaller than the value of the ensemble standard deviation over most of the global ocean. This suggests that some of the discrepancies between observations and models may be internally generated by the climate system.

We have incorporated this discussion into the revised text (Lines 124-126), and Fig. R8a has been added to Supplementary Fig. 4.

Figure R8 | Internal variabilities estimated as ensemble standard deviations. a-e, Ensemble standard deviation of linear trends of SST seasonal cycle amplitude among (a) CMIP6-MME, (b) ACCESS-ESM1-5 LENS, (c) CanESM5 LENS, (d) MIROC6 LENS, and (e) MPI-ESM1-2-LR LENS, with their zonal means shown in the right-hand panels. **f,** The difference between linear trends of SST seasonal cycle amplitude in OBS and EXT, with their zonal means shown in the right-hand panels. Stippling indicates where the difference is smaller than the ensemble standard deviation shown in (a).

L215 “Another intriguing feature is the widespread amplitude reduction beneath the mixed layer, attributed to the overall shallowing of the mixed layer that impedes the propagation of the seasonal cycle signal into deeper ocean layers.”: Interesting.

Response: Thank you. We are currently conducting further investigations to understand this phenomenon in observations. We are excited about the potential insights this may provide and look forward to sharing our findings in future research.

L242: “deoxygenation” rather than “oxygenation”?

Response: We apologize for the confusion, and have changed the term to "oxygen solubility" in the revised text accordingly.

L243 “The enhanced seasonal contrast of upper ocean oxygen level may foster the occurrence of hypoxic conditions characterized by exceedingly low oxygen concentrations by superimposing upon the long-term oxygen depletion trend over the past 60 years”: To quantify this impact, please show the spatial map of the relative changes in the sea surface oxygen amplitude (i.e., oxygen amplitude changes over the past 60 years divided by climatological seasonal amplitude).

Response: Thank you for your suggestion. Fig. R9 shows the relative changes in the seasonal amplitude of DO over the past 40 years. The strongest intensification appears to be in the subpolar North Atlantic (40-60°N, 60°W-0), with a relative change of over 15%. Intensification in the subpolar North Pacific (40-60°N, 140°E-120°W) and Southern Ocean (35-50°S) also reaches 5.5% and 6.9%, respectively. Note that the amplitude changes in the tropics (within 15°) are not shown because the small amplitude of the climatological annual cycle in these regions would generate extremely large ratio changes. Therefore, we have refrained from showing the ratio in the revision. However, we have incorporated the discussion into the revised text (Lines 259-262).

Figure R9 | Linear trend of surface dissolved oxygen seasonal cycle amplitude (unit: % per 40yr) during 1983-2022 from EXT. Stippling indicates where the trend is statistically significant above the 95% confidence level based on Student’s *t* test.

L325: "This calculation is based on the climatology of the period 1970-2000 from the ERSST": I could not get this. Please elaborate this.

Response: To clarify, we obtain the months of maximum and minimum SST based on the climatological period from 1970 to 2000 using the ERSST dataset, and these months serve as the basis for calculating the amplitude of the seasonal cycle for each year in both observations and simulations.

This clarification has been included in the revised text (Lines 343-346)

L338: "The MLD derived with the 0.01 kg m⁻³ threshold aligns perfectly with regions that exhibit an intensified seasonal cycle of upper ocean temperature at all latitudes": I think this is result and thus this is not placed in the "Method".

Response: Thank you, and we have reported this result in the text (Lines 225-226).

Figure 1a and b: The left y-axis should be "Monthly trend" rather than "Amplitude trend"?

Response: Corrected as suggested.

Figure S6 and Method: Why did not the authors use observational heat flux product (OAFflux <https://climatedataguide.ucar.edu/climate-data/oaflex-objectively-analyzed-air-sea-fluxes-global-oceans>, and JOFURO3 <https://www.j-ofuro.com/en/>)? Observational data is of limited duration, but should be used for validation of atmospheric reanalysis.

Response: We appreciate your suggestion to use observational heat flux products for validation purposes. However, upon careful examination, we found significant uncertainties in the estimation of SHF products, including discrepancies in the sign of the changes, as depicted in Fig. R10. These uncertainties in SHF estimation have been reported in previous studies (e.g., Large & Yeager, 2009; Yu et al., 2019), which makes it difficult to detect the long-term trends in the annual mean, let alone the amplitude of the SHF seasonal cycle.

Given the substantial uncertainties in the observational and reanalysis SHF datasets, and considering that climate models effectively capture the overall magnitude and spatial pattern of the intensification trend in the SST seasonal cycle, we rely primarily on CMIP6 simulations and a heat budget-based attribution analysis to reveal the contribution of SHF change to the intensification of the SST seasonal cycle in the revision (Lines 217-222).

Fig. R10 | Large uncertainties among surface heat flux products. a, The temporal evolution of the anomalies of the SHF (unit: W m^{-2}) averaged over 60°S and 60°N from observations, reanalysis, and climate model simulations. **b,** Same as a, but for the amplitude of the SHF seasonal cycle.

References:

Large, W. G., and S. G. Yeager, 2009: The global climatology of an interannually varying air–sea flux data set. *Climate Dynamics*, **33**, 341–364, <https://doi.org/10.1007/s00382-008-0441-3>.
 Yu, L., 2019: Global air-sea fluxes of heat, fresh water, and momentum: Energy budget closure and unanswered questions. *Annual Review of Marine Science*, **11**, 227–248, <https://doi.org/10.1146/annurev-marine-010816-060704>.

Figure S6b: This analysis also should be done for each hemisphere like Fig. S7.

Response: Thank you for your suggestion. In response to your comment, we have conducted a similar analysis for each hemisphere (Fig. R11), which shows a closer relationship between intensified amplitude in SHF and SST in the NH.

However, as we addressed in your major comment #2, correlation does not imply causality. Therefore, we have employed an attribution method to quantify the specific contribution of SHF amplitude changes to the intensified SST seasonal cycle. The results reveal that SHF amplitude changes have contributions in the North Pacific and some modest contributions in the southern subtropics (Fig. R6d). We have included this analysis in the revised manuscript (Lines 217-222).

Figure R11 | Inter-model relationship between the amplitude trend of SST seasonal cycle and surface heat flux seasonal cycle during 1982-2022 over the (a) northern hemisphere (0-60N) and (b) southern hemisphere (0-60S). The black line represents the linear regression line. Also shown is the correlation coefficient.

L176: What about the spatial correspondence between the heat flux changes and SST amplitude change?

Response: Instead of conducting a correlation analysis, we have employed a mixed-layer heat budget approach to directly evaluate the relationship between heat flux changes and SST amplitude changes. While the primary contribution of SHF amplitude changes is evident in the North Pacific (Fig. R6d), it is relatively small compared to the dominant role played by the decreasing MLD (Fig. R6a).

Data: Is there any reason why the authors did not use the CESM2 large ensemble (<https://esd.copernicus.org/articles/12/1393/2021/>)? The CESM2 is one of CMIP6 generation models and it has outputs from single forcing experiment (<https://journals.ametsoc.org/view/journals/clim/aop/JCLI-D-22-0666.1/JCLI-D-22-0666.1.xml>).

Response: The reason we did not include the CESM2 LENS data is that their future projection simulations are based on the SSP370 scenario rather than the SSP585 scenario, which we focused on for consistency in our analysis. However, it is worth noting that CESM2 LENS also shows seasonal amplitude intensification with a similar spatial pattern to that in EXT (Fig. R12).

Figure R12 | Linear trend of SST seasonal cycle amplitude (unit: °C per 40yr) during 1983-2022 from CEMS2 LENS. Note that the data over 2015-2022 is from the SSP3-7.0 scenario.

Method: 0.01 kg m⁻³ is too small for the threshold when determining the MLD from the Grid data product. As noted in the reference, 0.01 kg m⁻³ is a value to use for raw CTD profiles, rather 0.1 kg m⁻³ or 0.125 kg m⁻³ is more common because vertical profiles in gridded product are vertically smoothed.

Response: Thank you for your comments. We have considered various criteria for calculating the MLD, including the commonly used thresholds of 0.125 kg m⁻³ and 0.03 kg m⁻³, as recommended by Montégut et al. (2004) for defining MLD from vertical density profiles.

MLD estimates derived using the 0.03/0.125 kg m⁻³ criteria exhibit more pronounced increases in the southern subtropics and the southern flank of the ACC (Fig. R13b,c). However, the decreasing trends in MLD in the northern subpolar gyres and the northern flank of the ACC remain consistent under all three criteria (Fig. R13a-c), highlighting the robustness of the shallower MLD estimates in these regions.

We have chosen the 0.01 kg m⁻³ criterion due to its alignment with the intensification of upper ocean temperature seasonality (Fig R13d, blue line), suggesting that most of the seasonal cycle changes occur at this threshold. In contrast, the 0.125 kg m⁻³ criterion would yield significantly deeper MLD estimates in the high-latitude oceans (Fig. 13d, red line), where the seasonal cycle changes are relatively small.

Figure R13 | MLD calculated with different criteria. a-c, Linear trends of annual-mean MLD (unit: m per 40yr) during 1983-2022 from the IAP dataset calculated with (a) 0.01, (b) 0.03, and (c) 0.125 kg m⁻³ threshold. **d,** Linear trend of the zonal mean seasonal cycle of temperature (unit: m per 40yr) during 1983-2022 from the IAP dataset. The blue, orange, and red lines denote the annual mean MLD averaged over 1983-1992 calculated with 0.01, 0.03, and 0.125 kg m⁻³ threshold, respectively.

Reference:

de Boyer Montégut, C., 2004: Mixed layer depth over the global ocean: An examination of profile data and a profile-based climatology. *Journal of Geophysical Research*, 109, C12003, <https://doi.org/10.1029/2004JC002378>.

Figure S4a: It would be better to show the spatial map of the CMIP6 model spread (standard deviation) next to Fig. S4a.

Response: Thank you, and the standard deviation of the CMIP6 ensembles has been added as Supplementary Fig. 4b.

REVIEWERS' COMMENTS

Reviewer #1 (Remarks to the Author):

Thank you very much for addressing my questions and comments.
I am very happy to read the new and more insightful discussions.

Now, this work can be accepted and published by Nature Communications.

Reviewer #2 (Remarks to the Author):

Second review of Human-induced intensified seasonal cycle of sea surface temperature by Liu et al.

Thank you for addressing my concerns and for your sincere responses to the comments. I believe that the revised manuscript is much improved. In particular, the inclusion of the heat budget analysis performed by the authors has greatly clarified the discussion of the mechanism for the amplification of sea surface temperature. After addressing a few minor comments below, I can recommend this manuscript to the journal for publication.

Minor comments:

L224 and L228: Fig. 5b, d?

L281: The number seems not to be revised.

Fig. 1 caption: OBS is the mean of ERSST, HadISST, and OISST?

Fig. 4: Please clearly mention that this figure is a result from models (not observation) in the caption.

L373: When the authors calculate amplitudes of time-integrated tendency terms on the right-hand side of the equation, they use values on the months of maximum and minimum of T_m (not of time-integrated tendency terms), right? If so, please clarify it.

Supplementary Fig. 1a: While the caption says "OBS", the panel shows "OISST".

This review was also a valuable learning experience for me. Thank you.

Reviewed by Ryohei Yamaguchi

Note that major changes are highlighted in red in the revised text.

Reviewer #2 (Remarks to the Author):

Thank you for addressing my concerns and for your sincere responses to the comments. I believe that the revised manuscript is much improved. In particular, the inclusion of the heat budget analysis performed by the authors has greatly clarified the discussion of the mechanism for the amplification of sea surface temperature. After addressing a few minor comments below, I can recommend this manuscript to the journal for publication.

Thank you for your feedback and recommendation. We have addressed your comments to ensure clarity throughout the text.

Minor comments:

L224 and L228: Fig. 5b, d?

Response: Revised.

L281: The number seems not to be revised.

Response: Revised.

Fig. 1 caption: OBS is the mean of ERSST, HadISST, and OISST?

Response: Revised.

Fig. 4: Please clearly mention that this figure is a result from models (not observation) in the caption.

Response: Thank you, and we have revised as suggested.

L373: When the authors calculate amplitudes of time-integrated tendency terms on the right-hand side of the equation, they use values on the months of maximum and minimum of T_m (not of time-integrated tendency terms), right? If so, please clarify it.

Response: Yes, thank you for pointing this out. We have made it clear in the revision (Lines 374-376).

Supplementary Fig. 1a: While the caption says “OBS”, the panel shows “OISST”.

Response: The title of Supplementary Fig. 1a has been corrected.